Journal of Data-centric Machine Learning Research (2025)          Submitted 02/25; Revised 04/25; Published 06/25

# *MONSTER*
# Monash Scalable Time Series Evaluation Repository

**Angus Dempster**[*]                                                                ANGUS.DEMPSTER@MONASH.EDU
**Navid Mohammadi Foumani**[*]
**Chang Wei Tan**
**Lynn Miller**[*]
**Amish Mishra**[*]
**Mahsa Salehi**[*]
**Charlotte Pelletier**[†]
**Daniel F. Schmidt**[*]
**Geoffrey I. Webb**[*]
[*]*Monash University, Melbourne, Australia*
[†]*Université Bretagne Sud, IRISA, Vannes, France*

**Reviewed on OpenReview:** *https://openreview.net/forum?id=XauSqSfZfc*

**Editor:** Hugo Jair Escalante

## Abstract

We introduce MONSTER—the **MON**ash **S**calable **T**ime Series **E**valuation **R**epository—a collection of large datasets for time series classification and associated set of classification tasks that jointly define a new time series classification benchmark. The field of time series classification has benefitted from common benchmarks set by the UCR and UEA time series classification repositories. However, the datasets in these benchmarks are small, with median training set sizes of 217 and 255 examples, respectively. In consequence they favour a narrow subspace of models that are optimised to achieve low classification error on a wide variety of smaller datasets, that is, models that minimise variance, and give little weight to computational issues such as scalability. Our hope is to diversify the field by introducing benchmarks using larger datasets. We believe that there is enormous potential for new progress in the field by engaging with the theoretical and practical challenges of learning effectively from larger quantities of data.

**Keywords:** time series classification, dataset, benchmark, bitter lesson

## 1 Introduction

'State of the art' in time series classification has become synonymous with state of the art on the datasets in the UCR and UEA archives (Bagnall et al., 2018; Dau et al., 2019; Bagnall et al., 2017; Middlehurst et al., 2024; Ruiz et al., 2021). However, most of these datasets—at least, most of those that are commonly used for evaluation—are small: median training set size for the set of 142 canonical univariate time series datasets is just 217 examples. The preeminence of the datasets in the UCR and UEA archives as a basis for benchmarking

means that the field has become constrained by a narrow focus on smaller datasets and models which achieve low 0–1 loss (classification error) on a diversity of smaller datasets.

Empirical machine learning research relies heavily on benchmarking in one form or another (Liao et al., 2021). Benchmark datasets provide the data necessary for training and evaluating machine learning models. Certain datasets and benchmarks have become foundational to machine learning generally (Paullada et al., 2021). There is little doubt that the datasets in the UCR and UEA archives are as integral to the field of time series classification as are, for example, the MNIST, CIFAR, and ImageNet datasets to the field of image classification.

'[T]he ways in which we collect, construct, and share these datasets inform the kinds of problems the field pursues and the methods explored in algorithm development' (Paullada et al., 2021). We might call this the 'dataset lottery' or 'benchmark lottery'—after the 'hardware lottery'—i.e., to paraphrase Hooker (2021), when a method or set of methods 'win' (predominate) because of their compatibility with existing benchmarks.

A benchmark should serve as a proxy for a broader task (e.g., image classification, or time series classification). A given benchmark is only meaningful to the extent that performance on that benchmark reflects performance on a broader task, and performance on that benchmark generalises to real-world problems (Liao et al., 2021).

In the context of time series classification, current benchmarks favour models optimised to achieve low classification error (0–1 loss) on a diversity of smaller datasets, i.e., low-variance methods: see Section 2. Datasets currently used for benchmarking do not reflect either the theoretical or practical challenges of learning from large-scale real-world data.

This poses the risk that current benchmarks are unrepresentative of the broader task of time series classification, and that models considered state of the art on these benchmarks may not generalise to—and therefore may have diminishing relevance for—real-world time series classification problems, especially those involving larger quantities of data. This also suggests that research in time series classification has only so far explored a relatively narrow subset of ideas (see Hooker, 2021).

We present MONSTER—the **Mon**ash **S**calable **T**ime Series **E**valuation **R**epository—a collection of large univariate and multivariate datasets for time series classification. Our aim is to complement the existing datasets in the UCR and UEA archives, while encouraging the field to diversify to include significantly larger datasets. We hope that, with the introduction of MONSTER, benchmarking in the field better represents the broader task of time series classification, and has increased relevance for real-world time series classification problems. We hope to inspire the field to engage with the challenges of learning from large quantities of data. We believe that there is enormous potential for new progress in the field.

The rest of this paper is structured as follows. Section 2 expands on relevant background material. Section 3 provides further details of the MONSTER datasets. Section 4 provides preliminary baseline results for selected methods.

## 2 Motivation

### 2.1 Bias–Variance Tradeoff

A benchmark is useful in informing choices of learning algorithm for a new analytic task to the extent that the benchmark tasks are reflective of the task of interest. Whereas

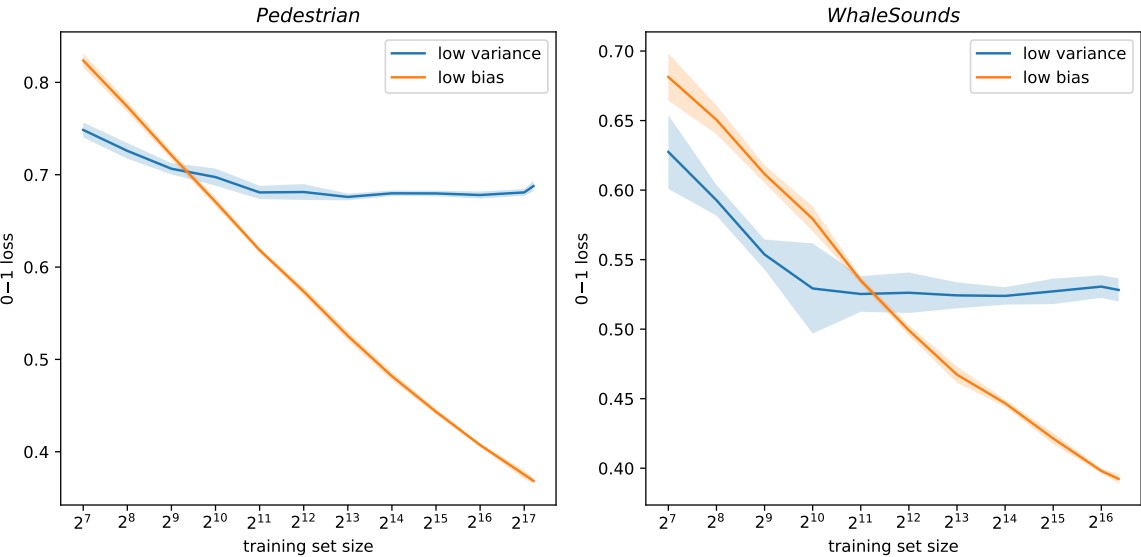

Figure 1: Learning curves (0–1 loss) for a low variance model (blue) versus a low bias model (orange) on the *Pedestrian* (left) and *WhaleSounds* (right) datasets. The shaded regions represent standard deviation over the cross validation folds.

historically, in the field of computer vision, different methods have generally been evaluated on benchmarks comprising a relatively small number of large datasets (e.g., ImageNet), in the field of time series classification, different methods are almost always evaluated on benchmarks comprising a relatively large number of small datasets, i.e., the datasets in the UCR and UEA archives.

With respect to a bias-variance decomposition of error (Sammut and Webb, 2017), the variance component of error (hereafter *variance*) can be expected to be large when training sets are small and to decrease as training set size increases. As a result, methods that effectively minimise variance will often achieve lower classification error on smaller datasets, while methods that minimise the bias component of error (hereafter simply *bias* where the context makes clear that we are referring to the bias component of error) will often achieve lower classification error on larger datasets (Brain and Webb, 1999). This is illustrated in Figure 1, which shows learning curves for two models: a low variance configuration of QUANT (a maximum tree depth of 4, with 128 trees) vs a low bias configuration of QUANT (unlimited tree depth, with 4 trees) on the *Pedestrian* and *WhaleSounds* datasets. Figure 1 shows that the low-variance model achieves lower 0–1 loss on smaller quantities of data, whereas the low-bias model achieves lower 0–1 loss on larger quantities of data. Additional results for these two models are provided in Section 4.6.

Note that minimizing bias on a learning task is more complex than simply choosing a learner that is inherently 'low bias.' The bias component of error when applying a learning algorithm is determined by the interaction between the form of the true classification function and assumptions about the classification function that are baked into the algorithm (the *inductive bias* of the algorithm). An algorithm with very strong inductive bias will

have a low bias component of error if those assumptions are fully satisfied, but a high bias component of error if they are not. For example, logistic regression will have a very low bias component of error if there is a linear relationship between the input variables and the log-odds of the class variable, and its bias component of error will increase proportionately to the extent to which this assumption is violated.

We should not expect the same methods to achieve the lowest 0–1 loss on both smaller datasets and larger datasets, as these demand different learning characteristics. (The issue of dataset size is not just limited to the quantity of training data: small quantities of test data can mask large differences in real-world classification error: Liao et al., 2021.)

As such, the methods currently considered state of the art in terms of accuracy on the datasets in the UCR and UEA archives are, by definition, likely dominated by methods optimised for smaller datasets or, in other words, methods that minimise variance. We see strategies for minimising variance in all or almost all state of the art methods for time series classification. Variance can be minimised via ensembling (e.g., InceptionTime (Ismail Fawaz et al., 2020), the HIVE-COTE models (Middlehurst et al., 2021), Proximity Forest (Lucas et al., 2019), and models such as DrCIF (Middlehurst et al., 2021) using ensembles of decision trees), explicit regularisation (e.g., methods using a ridge classifier such as RDST (Guillaume et al., 2022), Weasel 2.0 (Schäfer and Leser, 2023)), and/or overparameterisation (taking advantage of double descent, e.g., the ROCKET 'family' of methods (Dempster et al., 2020, 2021, 2023), and other methods making use of a large feature space in combination with a ridge regression classifier or other linear model, as well as large neural network models), or some combination of these approaches.

It is conceivable that this pressure to focus on controlling variance has directed the attention of researchers away from considerations of matching the inductive biases of learning algorithms to specific types of time series learning task. It is also conceivable, as discussed in Section 4.6, that the need to focus on controlling variance has directed attention away from the issues raised by concept shift that may be present in some benchmark tasks.

## 2.2 The 'Bitter Lesson'

It is not a coincidence that, with some exceptions, deep learning methods have had a relatively muted impact on the field. Models such as large deep neural networks are high variance models, and require significant quantities of training data in order to achieve competitive accuracy compared to less complex models. There has been a significant amount of work applying deep learning methods in the field of time series classification (Foumani et al., 2024a). However, despite this, and despite the fact that some neural network models such as InceptionTime (Ismail Fawaz et al., 2020) are among the most accurate models, on average, on the datasets in the UCR and UEA archives, in large part deep learning methods have not had the kind of impact that they have had in other domains such as image classification or natural language processing.

Arguably, time series classification has not yet had its 'ImageNet moment', simply because in almost all existing work the quantity of training data has been insufficient to allow for training low bias models such as large convolutional neural networks or transformer architectures effectively. (A not insubstantial amount of work involving deep learning in the

context of time series is also problematic, e.g., involving directly or indirectly optimising test loss: Middlehurst et al. (2024).)

It is not clear yet whether the 'bitter lesson'—'the only thing that matters in the long run is the leveraging of computation' (Sutton, 2019)—has yet been learned in the field of time series classification. The apparent diversity of methods considered state of the art may reflect a diversity of inductive biases that are effective for extracting information from low quantities of data, but that actually limit the ability to learn effectively from large quantities of data.

There is also the potential issue of overfitting a benchmark itself, although this is of less immediate concern due to the recent additional of new datasets to the UCR archive (Middlehurst et al., 2024). Accordingly, as well as being larger, the MONSTER datasets also represent new datasets or, in other words, a new 'out of sample' collection of datasets on which to evaluate existing methods.

### 2.3 'No Free Lunch'

Evaluation on a large set of heterogeneous datasets has led to another difference (in contrast to, e.g., computer vision), namely, that in the field of time series classification, performance is typically measured in terms of accuracy over all of the datasets in the UCR and/or UEA archives. This kind of average performance represents an average over a large set of highly heterogeneous input time series datasets.

This favours, without necessarily any good reason, methods that perform well (achieve low classification error) *on average*, while not necessarily performing well on any particular subset of datasets or tasks.

The 'no free lunch' theorem suggests that, as the number of datasets included in the evaluation grows, the performance of all methods should converge *on average*, i.e., no one method will perform better than any other on all datasets (Wolpert and MacReady, 1997). In the real world, this kind of average performance is potentially of limited practical value. For example, given a problem involving the classification of EEG data, we would rather use a method demonstrated to have good classification performance on benchmark EEG data, rather than a method that has low *average* classification error across both EEG data and data from one or more other domains.

In other words, current research likely unjustifiably favours methods that not only minimise variance, but that achieve low 0–1 loss *on average*, with potentially limited relevance to any specific real-world application.

In many cases it makes sense for a model or architecture to be specialised to a particular domain. For example, TempCNN uses short convolutional kernels, ideal for the short time series typical of Earth observation data, but which are not effective for capturing temporal relationships in long time series, e.g., those common in audio tasks. The lack of pooling layers allows TempCNN to locate temporal features important for tasks such as crop detection, but lacks the ability to detect scale-invariant features important in some other tasks (Pelletier et al., 2019). In contrast, ConvTran uses channel-wise convolutional kernels and attention to capture both relationships between channels and long-range temporal relationships, especially effective for EEG data (Foumani et al., 2024b), but which have potentially limited relevance to univariate and/or shorter time series.

## 2.4 Other Selection Pressures and the 'Hardware Lottery'

For the most part, the field has not been forced to contend with the practical challenges involved with learning from larger quantities of data. Just as smaller datasets favour methods that effectively minimise variance, different kinds of selection pressures exist in the context of larger datasets.

In particular, larger datasets select for methods that are computationally suited to large datasets, and can make effective use of existing computational resources, i.e., the 'hardware lottery' (Hooker, 2021). Methods with high computational and/or memory requirements quickly become impractical. Even for more efficient methods, training on large quantities of data presents significant engineering challenges.

## 2.5 Opportunities

The need for expanding benchmarking in the field to include larger datasets has been recognised for some time. Dau et al (2019) stated: '[p]erhaps a specialist archive of massive time series can be made available for the community in a different repository' (p 1295).

MONSTER represents an opportunity for the field to diversify to include large datasets, to engage with the challenges of learning from larger datasets, to better reflect the broader task of time series classification, and to improve relevance for real-world time series classification problems. We believe that there is enormous opportunity for new progress in the field.

Further, we make the following predictions in relation to the ways in which larger datasets might change the field of time series classification, which may or may not be borne out in practice in the long run:

- Only a subset of existing methods will be practical, i.e., those which can take advantage of current hardware to train efficiently.

- The methods which achieve the lowest 0–1 loss on larger datasets will differ from the methods which achieve the lowest 0–1 loss on smaller datasets.

- Average performance (e.g., average 0–1 loss) will become less relevant than performance within meaningful subsets of tasks (e.g., classification of EEG data, vs classification of satellite image time series data).

## 3 The MONSTER Datasets

The initial release of the MONSTER benchmark includes 28 univariate and multivariate datasets with between 10,299 and 59,268,823 time series. Table 1 provides an overview of the datasets. (We consider this as an initial release, and we aim to continue to add datasets to the benchmark.) The datasets are available via HuggingFace: `https://huggingface.co/monster-monash`. Additional information in relation to hosting is provided in Appendix B. Relevant code is available at: `https://github.com/Navidfoumani/monster`. We provide the datasets in `.npy` format to allow for ease of use with Python and straightforward memory mapping. (We also provide the datasets in legacy `.csv` format.) All datasets are under creative commons licenses or in the public domain, or we otherwise have been given permission to include the dataset in this collection. All datasets are already publicly available in some form.

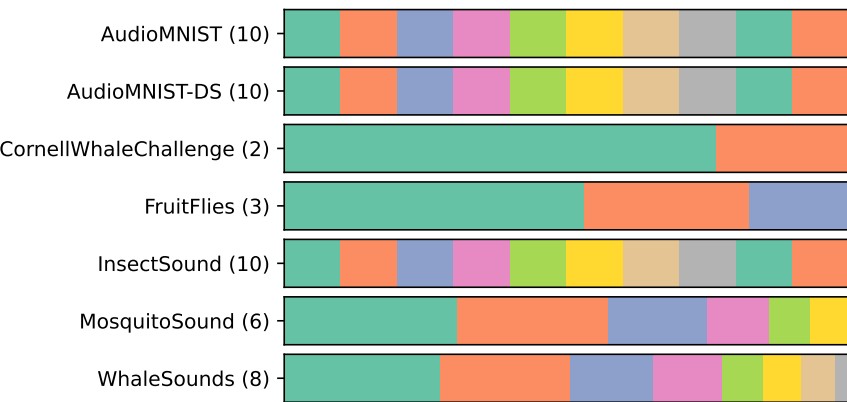

Figure 2: Class distributions for the audio datasets.

We have processed the original time series into a common format (`.npy` and `.csv`). The steps required to process each dataset were different and included, for example, extracting and labelling individual time series from broader time series data, interpolating irregularly sampled data, and resampling data where the original data was recorded at different sampling rates. We have endeavoured to lower the 'barrier of entry' as much as possible while keeping the original data intact to the greatest extent possible. Further details for each of the datasets are set out below.

Each dataset is provided with a set of indices for 5-fold cross-validation, allowing for direct comparison between benchmark results. For some datasets, these simply represent stratified random cross-validation folds. For other datasets, the cross-validation folds have been generated taking into account important metadata, e.g., different experimental subjects (for EEG data), or different geographic locations (for satellite image time series data). We have assigned the datasets to one of six categories (audio, satellite, EEG, HAR, count, and other). The distribution of classes for the datasets in each category is shown in Figures 2, 3, 6, 7, 10, and 11, below (in each figure, the number in brackets corresponds to the number of classes).

## 3.1 Audio

The learning tasks relating to audio data can range from classifying human speech to detecting insect species to identifying the presence of whales in hydrophone recordings. Audio data typically has a very high sampling rate (typically in kHz, or thousands of samples per second), but this can vary substantially depending on the exact application. The challenges involved in classifying audio data include both handling very long time series efficiently while extracting features at the appropriate resolution, as well as dealing with often large variability between different recordings.

### 3.1.1 AUDIOMNIST AND AUDIOMNIST-DS

**AudioMNIST** consists of audio recordings of 60 different speakers saying the digits 0 to 9, with 50 recordings per digit per speaker (Becker et al., 2024b,a). The speakers are a mixture

| Dataset | Instances | Length | SR | Channels | Classes |
|---|---:|---:|---:|---:|---:|
| | | Audio | | | |
| AudioMNIST | 30,000 | 47,998 | 48 kHz | 1 | 10 |
| AudioMNIST-DS | 30,000 | 4,000 | 4 kHz | 1 | 10 |
| CornellWhaleChallenge | 30,000 | 4,000 | 2 kHz | 1 | 2 |
| FruitFlies | 34,518 | 5,000 | 8 kHz | 1 | 3 |
| InsectSound | 50,000 | 600 | 6 kHz | 1 | 10 |
| MosquitoSound | 279,566 | 3,750 | 6 kHz | 1 | 6 |
| WhaleSounds | 105,163 | 2,500 | 250 Hz | 1 | 8 |
| | | Satellite Image Time Series | | | |
| LakeIce | 129,280 | 161 | daily | 1 | 3 |
| S2Agri | 59,268,823 | 24 | 10 days | 10 | 17 / 34 |
| S2Agri-10pc | 5,850,881 | 24 | 10 days | 10 | 17 / 29 |
| Tiselac | 99,687 | 23 | 16 days | 10 | 9 |
| | | EEG | | | |
| CrowdSourced | 12,289 | 256 | 128 Hz | 14 | 2 |
| DreamerA | 170,246 | 256 | 128 Hz | 14 | 2 |
| DreamerV | 170,246 | 256 | 128 Hz | 14 | 2 |
| STEW | 28,512 | 256 | 128 Hz | 14 | 2 |
| | | Human Activity Recognition | | | |
| Opportunity | 17,386 | 100 | 30 Hz | 113 | 5 |
| PAMAP2 | 38,856 | 100 | 100 Hz | 52 | 12 |
| Skoda | 14,117 | 100 | 98 Hz | 60 | 11 |
| UCIActivity | 10,299 | 128 | 50 Hz | 9 | 6 |
| USCActivity | 56,228 | 100 | 100 Hz | 6 | 12 |
| WISDM | 17,166 | 100 | 20 Hz | 3 | 6 |
| WISDM2 | 149,034 | 100 | 20 Hz | 3 | 6 |
| | | Counts | | | |
| Pedestrian | 189,621 | 24 | hourly | 1 | 82 |
| Traffic | 1,460,968 | 24 | hourly | 1 | 7 |
| | | Other | | | |
| FordChallenge | 36,257 | 40 | 10 Hz | 30 | 2 |
| LenDB | 1,244,942 | 540 | 20 Hz | 3 | 2 |

Table 1: Summary of Monster datasets.

of ages and genders. The recordings are single channel have a sampling rate of 48 kHz. The learning task is to classify the spoken digit based on the audio recording. The processed dataset contains 30,000 (univariate) time series, each of length 47,998 (approximately 1 second of data sampled at 48 kHz), with ten classes representing the digits 0 to 9. This version of the dataset has been split into cross-validation folds based on speaker (i.e., such that recordings for a given speaker do not appear in both the training and validation sets). ***AudioMNIST-DS*** is a variant of the same dataset where the time series have been downsampled to a length of 4,000 (i.e., effectively 4 kHz).

### 3.1.2 CORNELLWHALECHALLENGE

***CornellWhaleChallenge*** consists of hydrophone recordings (Karpištšenko et al., 2013). The recordings are single channel with a sampling rate of 2 kHz. The recordings come from an array of buoys near Boston. The processed dataset consists of 30,000 (univariate) time series, each of length 4,000 (i.e., representing recordings of 2 seconds of audio with a sampling rate of 2 kHz). The task is to distinguish right whale calls from other noises. (An abridged version of this dataset is included in the broader UCR archive.) This version of the dataset has been divided into stratified random cross-validation folds.

### 3.1.3 FRUITFLIES

***FruitFlies***, taken from the broader UCR archive, consistst of 34,518 (univariate) time series, each of length 5,000, representing acoustic recordings of wingbeats for three species of fruit fly (Potamitis, 2016; Flynn, 2022). The recordings are single channel with a sampling rate of 8 kHz (i.e., each recording represents just over half a second of data). The recordings are made using a specialised infrared sensor which detects the vibrations of the wings of the insects. The learning task is to identify the species of fly based on the recordings. This version of the dataset has been split into stratified random cross-validation folds.

### 3.1.4 INSECTSOUND

***InsectSound***, taken from the broader UCR archive, consists of 50,000 (univariate) time series, each of length 600, representing recordings of wingbeats for six species of insects, with 2 different genders for 4 of the 6 species (Chen et al., 2014; Chen, 2014). The recordings are single channel with a sampling rate of 6 kHz (i.e., each time series represents 10 ms of data). Similar to *FruitFlies*, but using different hardware, the recordings were made using an infrared sensor detecting the vibrations of the wings of the insects. The learning task is to identify the species of insect based on the recordings. This version of the dataset has been split into stratified random cross-validation folds.

### 3.1.5 MOSQUITOSOUND

***MosquitoSound***, taken from the broader UCR archive, consists of 279,566 (univariate) time series, each of length 3,750, representing recordings of wingbeats for six different species of mosquito (Fanioudakis et al., 2018; Potamitis, 2018). The recordings are single channel with a sampling rate of 6 kHz (i.e., the time series represent just over half a second of data). As for the *FruitFlies* dataset, and using similar hardware, the recordings were made using

an infrared sensor detecting the vibration of the wings of the mosquitoes. The task is to identify the species of mosquito based on the recordings. This version of the dataset has been split into stratified random cross-validation folds.

### 3.1.6 WHALESOUNDS

**WhaleSounds** consists of underwater acoustic recordings around Antarctica, manually annotated for seven different types of whale calls (Miller et al., 2020, 2021). The recordings are single channel. The original data consists of extended recordings with a mixture of different sampling rates between 250 and 2,500 Hz. The dataset has been processed to extract the annotated whale calls from the original recordings. The extracted whale calls have been resampled to a consistent sampling frequency of 250 Hz. (The whale sounds are typically well below 100 Hz.) The processed dataset contains 105,163 (univariate) time series, each of length 2,500 (i.e., each time series represents 10 seconds of data at 250 Hz, approximately centred on the labelled whale sound), with eight classes representing the seven types of whale call plus a class for unidentified sounds. This version of the dataset has been split into stratified random cross-validation folds.

## 3.2 Satellite Image Time Series

Satellite image time series consist of data recorded over time at a particular location derived from images captured by sensors on board Earth observation satellites. Satellite images taken over time (e.g., every five days) at the same location produce time series at a pixel level. Each time series represents changing values for a given pixel over time. The different channels represent the different spectral bands captured by the given satellite (for sensors covering the visible and infrared frequencies) or the polarisation of the microwave signal (for microwave sensors). The satellites used to collect the data used in the MONSTER datasets follow near-polar, low-Earth orbits. These satellites orbit the Earth approximately every 90 minutes and their orbital path means they can image nearly all the Earth's surface over a few days (10 to 16 days, depending on the satellite). However, this leads to a relatively low sampling frequency at any given location. Given this low sampling frequency, satellite image time series are often relatively short, and these datasets often have strong temporal alignment (the time series cover the same time period, or the same time period in different years, and typically with the same or similar sampling dates). The key challenges for satellite image time series are handling potentially very large volumes of data (even relatively low resolution satellite imagery over the whole surface of the earth corresponds to trillions of time series), and differing patterns corresponding to different geographic locations, geographies, and climatic conditions.

### 3.2.1 LAKEICE

**LakeIce** consists of pixel-level backscatter (reflection) values from satellite images of an area of approximately 6,000 km$^2$ in Yukon, Canada (Shaposhnikova et al., 2022, 2023). The time series are extracted over three decades from ERS-1/2, Radarsat, and Sentinel-1 synthetic aperture radar satellites, which all use the C-band range of microwave frequencies (4-8GHz). This is a pixel-level dataset, such that each time series represents values over time for single pixel. The satellites used in this case have different spatial resolutions, resulting in a mixture

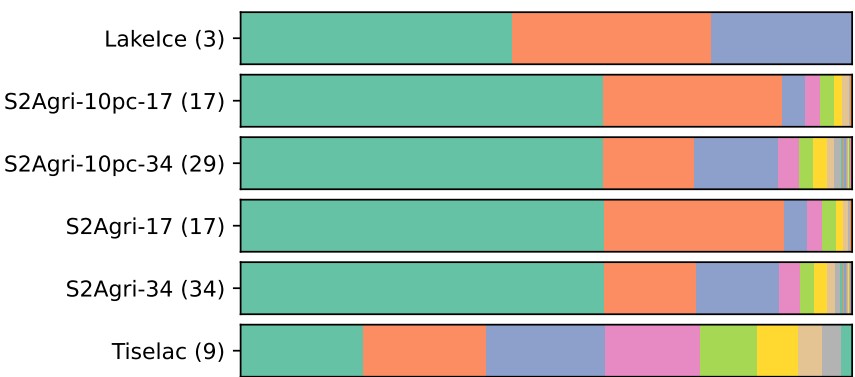

Figure 3: Class distributions for the satellite datasets.

of effective pixel sizes of between 12.5m, 30m, and 50m. The processed dataset contains 129,280 (univariate) time series each of length 161, representing daily data over near 6 months (October to March), with three classes, labelled manually, representing bedfast ice, floating ice, and land. (The original data has been calibrated and speckle-filtered, and then interpolated to provide daily values for the relevant period (Shaposhnikova et al., 2023).) This version of the dataset has been split into stratified random cross-validation folds.

### 3.2.2 S2AGRI

**S2Agri** is a land cover classification dataset and contains a single tile of Sentinel-2 data (T31TFM), which covers a 12,100 km2 area in France: see Figure 4 (Garnot et al., 2020; Sainte Fare Garnot and Landrieu, 2022). Ten spectral bands covering the visible and infrared frequencies are used, and these are provided at 10m resolution. The dataset contains time series of length 24, observed between January and October 2017, with data sampled approximately every 10 days. The area has a wide range of crop types and terrain conditions.

The original S2Agri dataset is designed for parcel-based processing and contains data for 191,703 land parcels, with data for each parcel provided in a separate file. We have reorganised the data for pixel-based processing, leading to a dataset containing 59,268,823 pixels. Two sets of land cover classification labels are provided, one with 19 classes and the other with 44 classes. However, some of the 44-classes are only represented by one land parcel. We have removed the pixels in these land parcels from the dataset. This means there are only 17 and 34 classes respectively that are represented in the final dataset. The class label of each parcel comes from the French Land Parcel Identification System. The dataset is unbalanced: the largest four of the 19-class labels account for 90% of the parcels.

We thus provide two versions of the S2Agri dataset, **S2Agri-17**, which uses the 17 class labels and **S2Agri-34**, which uses the 34 class labels. Additionally, we have created smaller versions of the datasets consisting of data for a randomly selected 10% of the land parcels, each containing 5,850,881 pixels. These are **S2Agri-10pc-17** and **S2Agri-10pc-34** for the 17-class and 34-class labels, respectively.

To create the folds used for cross-validation, we split the data based on the original land parcels, thus ensuring that all pixels in a land parcel are allocated to the same fold. Splits are stratified by class labels to ensure an even representation of the classes across the folds.

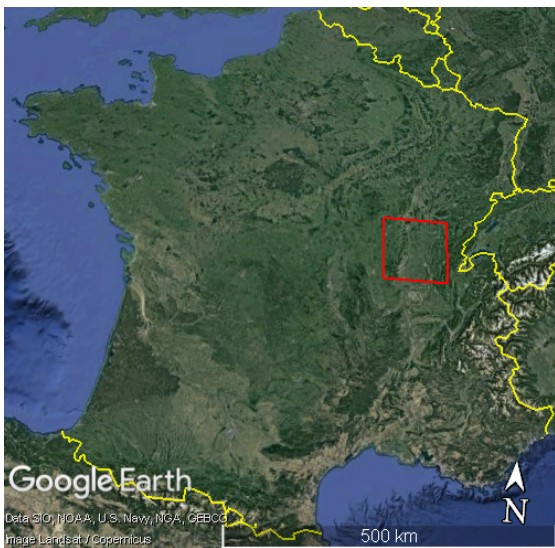

Figure 4: Map of France showing the location of the Sentinel-2 tile used in *S2Agri*.

### 3.2.3 TiSeLaC

***TiSeLaC*** (Time Series Land Cover Classification) was created for the time series land cover classification challenge held in conjunction with the 2017 European Conference on Machine Learning & Principles and Practice of Knowledge Discovery in Databases (Ienco, 2017). It was generated from a time series of 23 Landsat 8 images of Reunion Island (Figure 5a), sampled approximately every 16 days, acquired in 2014. This is a pixel level dataset, where each time series represents changing values for a single pixel. Ten time series features are provided for each pixel, seven surface reflectances covering visible and infrared frequencies and three indices derived from these bands: the Normalised Difference Vegetation Index, the Normalised Difference Water Index, and the Brightness Index. At the 30m spatial resolution of Landsat 8 images, Reunion Island is covered by $2866 \times 2633$ pixels, however only 99,687 of these pixels are included in the dataset. Class labels were obtained from the 2012 Corine Land Cover (CLC) map and the 2014 farmers' graphical land parcel registration (Régistre Parcellaire Graphique or RPG) and the nine most significant classes have been included in the dataset. The number of pixels in each class is capped at 20,000. The data was obtained from the winning entry's GitHub repository (Di Mauro et al., 2017), which includes the spatial coordinates for each pixel. The processed dataset consists of $99,687$ multivariate time series each of length 23 (i.e., representing approximately one year of data per time series at a sampling period of approximately 16 days).

We provide training and testing splits designed to give spatial separation between the splits, which should lead to realistic estimations of the generalisation capability of trained models. We first divided the original pixel grid into a coarse grid, with each grid cell sized at $300 \times 300$ pixels, then computed the number of dataset pixels in each cell (the cell size). These cells are processed in descending order of size, and allocated to the fold with the fewest pixels. The resulting spatial distribution of the folds is shown in Figure 5a and the distribution of classes across the folds is shown in Figure 5b.

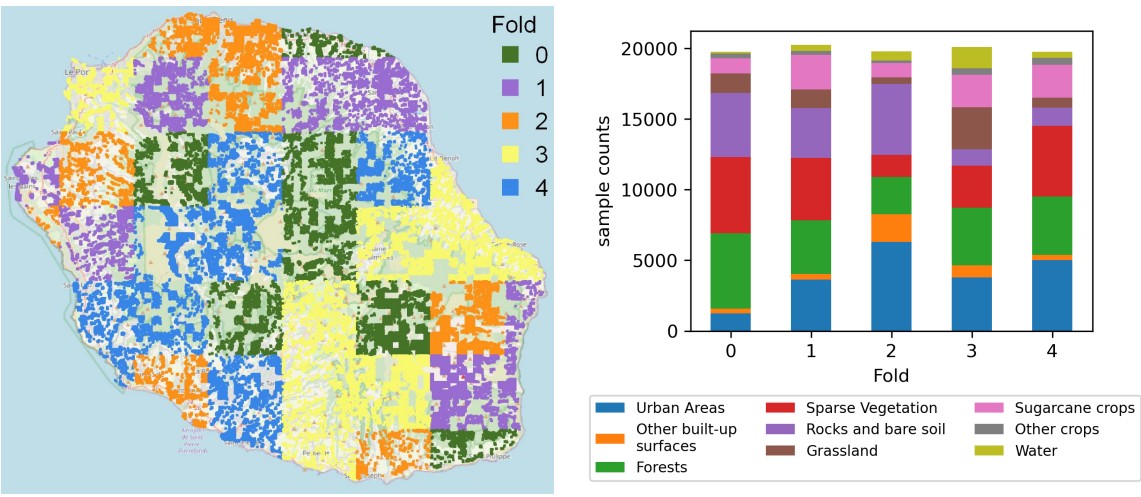

(a) Map of Reunion Island and fold data distribution  (b) Label counts by fold

Figure 5: Map of Reunion Island and label counts by fold for the Tiselac dataset. (Map from Open Street Map, sample data pixels are not to scale.)

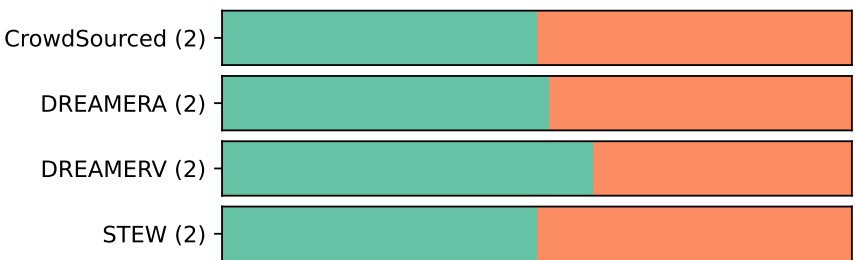

Figure 6: Class distributions for the EEG datasets.

### 3.3 EEG

An electroencephalogram (EEG) is a non-invasive method that captures brain activity by placing electrodes on the surface of the scalp, allowing for the recording of electrical signals generated within the brain. EEG data is typically recorded at high sampling rates (hundreds of samples per second) and is widely used in tasks such as classifying cognitive states or detecting neurological conditions. Despite these benefits, EEG analysis presents several challenges. A key issue is that EEG recordings often capture a mixture of signals, not just from the brain but also from other sources (i.e., noise). Each electrode records cortical activity along with non-cortical signals, such as muscle movements, and even environmental noise, like electrical interference. Furthermore, EEG data can vary significantly both across individuals (inter-subject variability) and within the same individual across different sessions (intra-subject variability).

### 3.3.1 CROWDSOURCED

***CrowdSourced*** consists of EEG data collected as part of a study investigating brain activity during a resting state task, which included two conditions: *eyes open* and *eyes closed*, each lasting 2 minutes. The dataset contains EEG recordings from 60 participants, but only 13 successfully completed both conditions. The recordings were captured using 14-channel EEG headsets—specifically the *Emotiv EPOC+*, *EPOC X*, and *EPOC* devices. These devices provide high-quality, wireless brainwave data that is ideal for analyzing resting-state brain activity (Williams et al., 2023).

The data was initially recorded at a high frequency of 2048 Hz and later downsampled to 128 Hz for processing. To segment the data for analysis, we used a 2-second window (equivalent to 256 time steps) with a 32 time-step stride to capture the dynamics of brain activity while maintaining a manageable data size. The raw EEG data for the 13 participants, along with preprocessing steps, analysis scripts, and visualization tools, are openly available on the Open Science Framework (Williams et al., 2022). The processed dataset consists of 12,289 multivariate time series, each of length 256 (i.e., representing 2 seconds of data per time series at a sampling rate of 128 Hz). This version of the dataset has been split into cross-validation folds based on participant.

### 3.3.2 DREAMERA AND DREAMERV

***Dreamer*** is a multimodal dataset that includes electroencephalogram (EEG) and electrocardiogram (ECG) signals recorded during affect elicitation using audio-visual stimuli (Katsigiannis and Ramzan, 2017b), captured with a 14-channel Emotiv EPOC headset at a sampling rate of 128 Hz. It consists of data recorded from 23 participants, along with their self-assessments of affective states (valence, arousal, and dominance) after each stimulus (Katsigiannis and Ramzan, 2017b). For our classification task, we focus on the arousal and valence labels, referred to as ***DreamerA*** and ***DreamerV*** respectively. The processed datasets both consist of 170,246 multivariate time series each of length 256 (i.e., representing 2 seconds of data per time series at a sampling rate of 128 Hz).

The dataset is publicly available (Katsigiannis and Ramzan, 2017a), and we utilize the Torcheeg toolkit for preprocessing, including signal cropping and low-pass and high-pass filtering (Zhang et al., 2024). Note that only EEG data is analyzed in this study, with ECG signals excluded. Labels for arousal and valence are binarized, assigning values below 3 to class 1 and values of 3 or higher to class 2, and has been split into cross-validation folds based on participant.

### 3.3.3 STEW: SIMULTANEOUS TASK EEG WORKLOAD

***STEW*** comprises raw EEG recordings from 48 participants involved in a multitasking workload experiment (Lim et al., 2018). Additionally, the subjects' baseline brain activity at rest was recorded before the test. The data was captured using the Emotiv Epoc device with a sampling frequency of 128Hz and 14 channels, resulting in 2.5 minutes of EEG recording for each case. Participants were instructed to assess their perceived mental workload after each stage using a rating scale ranging from 1 to 9, and these ratings are available in a separate file. The dataset has been divided into cross-validation folds based on individual participants. Additionally, binary class labels have been assigned to the data, categorizing

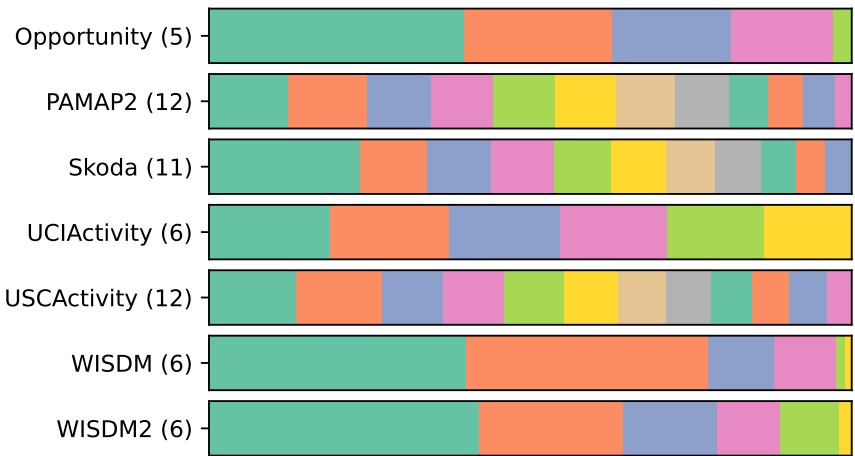

Figure 7: Class distributions for the HAR datasets.

workload ratings above 4 as "high" and those below or equal to 4 as "low". We utilize these labels for our specific problem. STEW can be accessed upon request through the IEEE DataPort (Lim et al., 2020). The processed dataset consists of 28,512 multivariate time series each of length 256 (i.e., representing 2 seconds of data at 128 Hz).

## 3.4 Human Activity Recognition

Human activity recognition (HAR) time series consist of data recorded over time from various sensors placed on the body. The rise of wearable technologies and the Internet of Things (IoT) has led to a significant increase in activity data collection, enabling widespread applications aimed at enhancing safety and quality of life in fields such as healthcare, fitness monitoring, smart homes, and assisted living. HAR data is typically captured at high sampling rates (multiple samples per second) using a variety of sensors, with wearable sensors being the most common. These sensors include smartphones, motion sensors, and other embedded devices. The primary task in HAR is classifying subjects' activities based on sensor readings. Similar to EEG time series data, the key challenges involved in classifying HAR time series data include various sources of noise, as well as potentially large differences between experimental subjects.

### 3.4.1 OPPORTUNITY

**Opportunity** is a comprehensive, multi-sensor dataset designed for human activity recognition in a naturalistic environment (Chavarriaga et al., 2013). Collected from four participants performing typical daily activities, the dataset spans six recording sessions per person: five unscripted "Activities of Daily Living" (ADL) runs, and one structured "drill" run with specific scripted activities. This dataset includes rich, multi-level annotations; however, for our analysis, we focus specifically on the locomotion classes, which consist of five primary categories: Stand, Walk, Sit, Lie, and Null (no specific activity detected).

Data collection includes 113 sensor channels from body-worn, object-attached, and ambient sensors with a sampling rate of 30 Hz. These channels capture detailed information

on body movements, object interactions, and environmental context through a combination of 7 inertial measurement units (IMUs), 12 3D accelerometers, 4 3D localization sensors, 12 object-attached 3D accelerometers with 2D rate-of-turn sensors, 13 switches, and 8 ambient 3D accelerometers. The variety and placement of these sensors allow for detailed examination of physical activities and transitions in a natural setting. To prepare the data for analysis, we segment it using a sliding window approach with a 100 time-step window and an overlap of 50 time steps. This segmentation enables the model to capture both the continuity of activities and subtle transitions, enhancing recognition accuracy across the locomotion classes. The processed dataset consists of 17,386 multivariate time series each of length 100 (i.e., representing just over 3 seconds of data per time series at 30 Hz). The dataset has been divided into cross-validation folds based on individual participants.

### 3.4.2 PAMAP2: Physical Activity Monitoring Dataset

**PAMAP2** is a collection of data obtained from three Inertial Measurement Units (IMUs) placed on the wrist of the dominant arm, chest, and ankle, as well as 1 ECG heart rate (Reiss and Stricker, 2012). The data was recorded at a frequency of 100Hz. The dataset includes annotated information about human activities performed by 9 subjects, each with their own unique physical characteristics. The majority of the subjects are male and have a dominant right hand. Notably, the dataset includes only one female subject (ID 102) and one left-handed subject (ID 108). In total, there are 12 different human activity classes represented in the dataset. The processed dataset contains 38,856 time series each of length 100 (i.e., representing one second of data per time series at 100 Hz). To ensure an unbiased evaluation, we divide the dataset into cross-validation folds based on the subjects.

### 3.4.3 Skoda: Mini Checkpoint-Activity recognition dataset

**Skoda** captures 10 specific manipulative gestures performed in a car maintenance scenario (Zappi et al., 2012). Its purpose is to investigate different aspects related to the gestures, such as fault resilience, performance scalability with the number of sensors, and power performance management. The dataset comprises 10 classes of manipulative gestures, which were recorded using $2 \times 10$ USB 3D acceleration sensors positioned on the left and right upper and lower arm. The sensors have a high sample rate of approximately 98 Hz, ensuring precise capturing of the movements.

In terms of activities, the dataset includes 10 distinct manipulative gestures commonly performed during car maintenance (Figure 8). The data was collected from a single subject, with each gesture being recorded 70 times. In total, the dataset offers around 3 hours of recording time, enabling thorough analysis of the gestures in various scenarios. The processed dataset consists of 14,117 time series each of length 100 (i.e., representing approximately one second of data per time series at 98 Hz).

### 3.4.4 UCIActivity

**UCIActivity** is a widely recognized benchmark for activity recognition research. It contains sensor readings from 30 participants performing six daily activities: walking, walking upstairs, walking downstairs, sitting, standing, and lying down. The data was collected using a Samsung Galaxy S2 smartphone mounted on the waist of each participant, recording

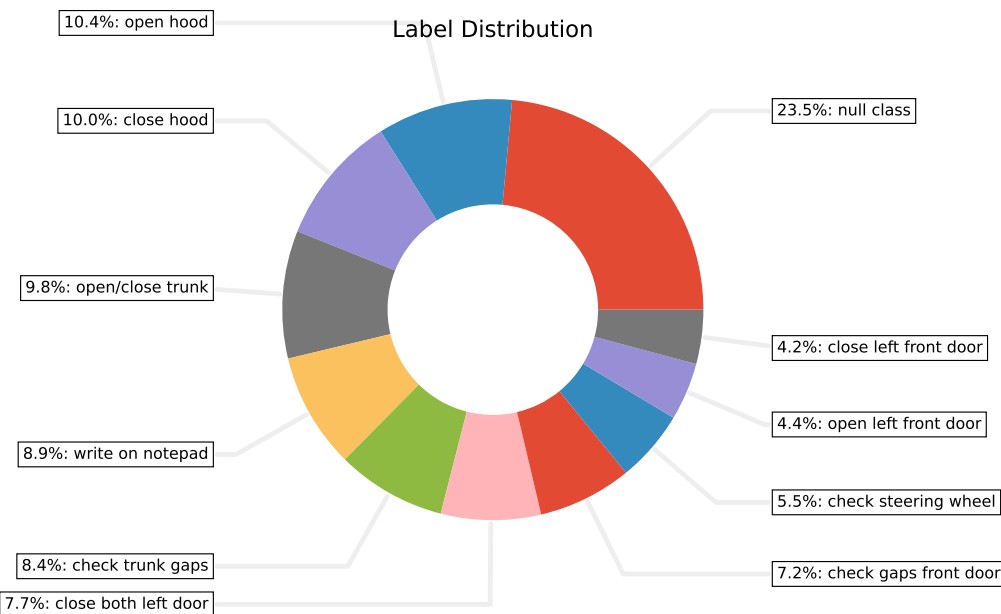

Figure 8: Distribution of activity categories for *Skoda*.

9 channels of data, with a sampling rate of 50 Hz (Anguita et al., 2013). The processed dataset contains 10,299 multivariate time series each with length 50 (i.e., one second of data at a sampling rate of 50 Hz). To keep the evaluation fair, we perform subject-wise cross-validation.

### 3.4.5 USCACTIVITY: USC HUMAN ACTIVITY DATASET

**USCActivity** (Zhang and Sawchuk, 2012) consists of data collected from a Motion-Node device, which includes six readings from a body-worn 3-axis accelerometer and gyroscope sensor. The dataset contains samples from 14 male and female subjects with equal distribution (7 each) and specific physical characteristics and ages. The sensor data is sampled at a rate of 100 Hz, and each time-step in the dataset is labelled with one of 12 activity classes (Figure 9). The processed dataset consists of 56,228 multivariate time series each of length 100 (representing one second of data at 100 Hz).

The USCActivity dataset presents a challenge in learning feature representation and segmentation due to the placement of the sensors and the variability in activity classes. The data is collected from a single accelerometer and gyroscope reading obtained from a motion node attached to the subject's right hip. Therefore, this reading does not contribute significantly to the feature space transformation. Additionally, the activity classes involve various orientations, such as walking forward, left, or right, and even using the elevator up or down, which cannot be captured solely through accelerometer and gyroscope readings. Similar to other activity recognition datasets, we use subject-based cross-validation.

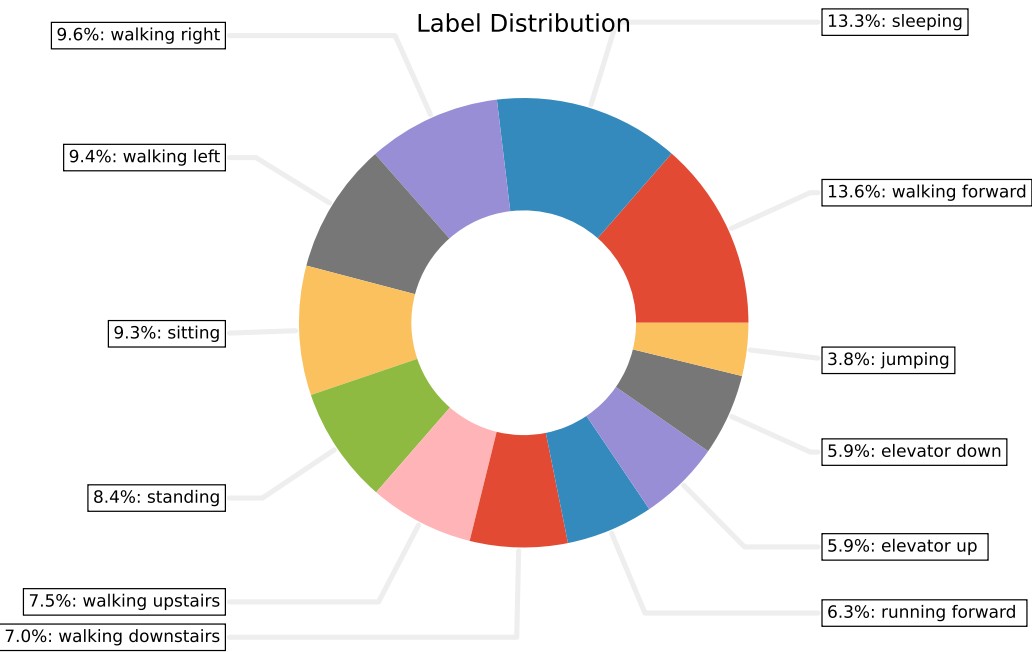

Figure 9: Distribution of activity categories for *USCActivity*.

### 3.4.6 WISDM AND WISDM2: WIRELESS SENSOR DATA MINING

**WISDM** describes six daily activities—*Walking*, *Jogging*, *Stairs*, *Sitting*, *Standing*, and *Lying Down*—collected in a controlled laboratory environment. Data were recorded from 36 participants using a smartphone's built-in tri-axial accelerometer, with the device placed in the user's front pants pocket. The accelerometer captures acceleration along the x, y, and z axes, providing a comprehensive view of the user's movements. The data is sampled at a rate of 20 Hz, resulting in a total of 1,098,207 samples across 3 dimensions (Lockhart et al., 2012). The processed dataset contains 17,166 multivariate time series with a length of 100 (representing 5 seconds of data at 20 Hz).

**WISDM2** extends the original *WISDM* dataset by collecting data in real-world environments using the Actitracker system. This system was designed for public use and provides a more extensive collection of sensor readings from users performing the same six activities. The dataset contains 2,980,765 samples with three dimensions, and the data was recorded from a larger and more diverse set of participants in naturalistic settings, offering a valuable resource for real-world activity recognition (Weiss and Lockhart, 2012). The processed dataset has 149,034 time series, each with length 100 (again, representing 5 seconds of data at a sampling rate of 20 Hz). Both WISDM and WISDM2 are split based on subjects.

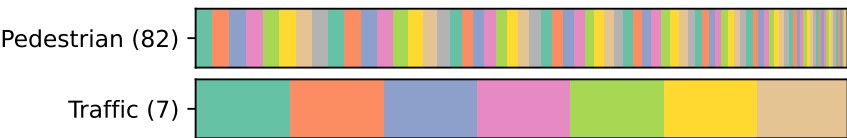

Figure 10: Class distributions for the count datasets.

## 3.5 Count

These datasets consists automatic sensor count data recorded over time. Depending on sampling frequency, count data can be aggregated at different resolutions (e.g., per minute, per hour, or per day), and for various different durations (e.g., hourly counts over a day, versus daily counts over a year). Given the nature of the data, time series of counts tend to have strong temporal alignment (e.g., different time series of counts over a 24-hour period all begin and end at the same time of day, with the same sampling frequency). There are various challenges involved in classifying count data, for example, variability in the patterns of counts over different periods (e.g., on different days of the week, and as patterns change over longer periods of time), and differences in the patterns at different locations.

### 3.5.1 PEDESTRIAN

**Pedestrian** represents hourly pedestrian counts at 82 locations in Melbourne, Australia between 2009 and 2022 (City of Melbourne, 2022). The processed dataset consists of 189,621 (univariate) time series, each of length 24 (i.e., representing 24 hours of data per time series). The data comes from automatic pedestrian counting sensors at different locations. The task is to identify location based on the time series of counts. The dataset has been split into stratified random cross-validation folds.

### 3.5.2 TRAFFIC

**Traffic** consists of hourly traffic counts at various locations in the state of NSW, Australia (Transport for NSW, 2023). The processed dataset contains 1,460,968 (univariate) time series, each of length 24 (i.e., representing 24 hours of data per time series). The data comes from automatic traffic counting sensors at different locations. The task is to predict the day of the week based on the time series of counts. The dataset has been split into stratified random cross-validation folds.

## 3.6 Other

Two datasets, *FordChallenge* and *LenDB*, do not neatly fall into one of the other categories, and represent distinct learning tasks compared to other datasets. (We anticipate adding additional categories as the benchmark is expanded over time.) *FordChallenge* represents data recorded over time from a variety of different sensors while an experimental subject is driving a car. *LenDB* contains seismological data recorded from seismic monitoring stations.

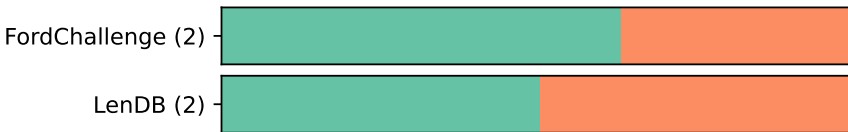

Figure 11: Class distributions for the uncategorised datasets.

### 3.6.1 FORDCHALLENGE

***FordChallenge*** is obtained from Kaggle and consists of data from 600 real-time driving sessions, each lasting approximately 2 minutes and sampled at 100ms intervals (Abou-Nasr, 2011) (i.e., a sampling rate of 10 Hz). The processed dataset consists of 36,257 multivariate time series each of length 40 (i.e., representing 4 seconds of data per time series at 10 Hz). These sessions include trials from 100 drivers of varying ages and genders. The dataset contains 8 physiological, 11 environmental, and 11 vehicular measurements, with specific details such as names and units undisclosed by the challenge organizers. Each data point is labeled with a binary outcome: 0 for "distracted" and 1 for "alert." The objective of the challenge is to design a classifier capable of accurately predicting driver alertness using the provided physiological, environmental, and vehicular data.

### 3.6.2 LENDB

***LenDB*** consists of seismograms recorded from multiple different seismic detection networks from across the globe (Magrini et al., 2020a,b). The sampling rate is 20 Hz. The processed dataset consists of 1,244,942 multivariate time series, with 3 channels, each of length 540 (i.e., just under 30 seconds of data per time series at a sampling rate of 20 Hz), with two classes: earthquake and noise. This version of the dataset has been split into cross-validation folds based on seismic detection network (i.e., such that seismograms for a given network do not appear in both a training and validation fold).

## 4 Baseline Results

### 4.1 Models

We provide baseline results on the MONSTER datasets for a number of key models. In particular, we provide results for four deep learning models: ConvTran (Foumani et al., 2024b), FCN (Wang et al., 2017), HInceptionTime (Ismail-fawaz et al., 2022), and TempCNN (Pelletier et al., 2019). We include results for two more 'traditional', specialised methods for time series classification: HYDRA (Dempster et al., 2023), and QUANT (Dempster et al., 2024a). We also include results for a standard, 'off the shelf' classifier—extremely randomised trees (Geurts et al., 2006)—to act as a naïve baseline.

**FCN** is a fully convolutional neural network. It consists of three temporal convolutional layers (one-dimensional convolutional layers that convolve along the time series), followed by a global average pooling layer and finally the softmax classification layer (Wang et al., 2017). The convolutional layers have 128, 256, and 128 filters of length 8, 5, and 3, respectively.

**TempCNN** is a light-weight temporal convolutional neural network originally designed for land cover classification from time series of satellite imagery (Pelletier et al., 2019). It

consists of three temporal convolutional layers followed by a fully connected layer. Each convolutional layer has 64 filters of length 5 and the fully-connected layer has 256 units.

**H-InceptionTime** (Hybrid-InceptionTime) is an ensemble of five Hybrid-Inception (H-Inception) models, each with a different random weight initialisation (Ismail-fawaz et al., 2022). An H-Inception model consists of a set of 17 hand-crafted filters combined with six Inception modules. The hand-crafted filters are sets of convolutional filters designed to detect peaks, and both increasing and decreasing trends. The hand-crafted filters range in length from 2 to 96 and are applied in parallel with the first inception module to the input time series. Inception modules combine convolutions with filter lengths of 10, 20 and 40, max pooling and bottleneck layers. Each set of convolutions and the max pooling layer have 32 filters thus each inception module has 128 filters. The resulting network has a small number of parameters and a large receptive field (Ismail Fawaz et al., 2020).

**ConvTran** is a deep learning model for multivariate time series classification (TSC) that combines convolutional layers with transformers to effectively capture both local patterns and long-range dependencies (Foumani et al., 2024b). It addresses the limitations of existing position encoding methods by introducing two novel techniques: tAPE (temporal Absolute Position Encoding) for absolute positions and eRPE (efficient Relative Position Encoding) for relative positions. These encodings, integrated with disjoint temporal and channel-wise convolutions (Foumani et al., 2021), allow ConvTran to capture both temporal dependencies and correlations between the channels.

**Hydra** involves transforming input time series using a set of random convolutional kernels arranged into groups, and 'counting' the kernel representing the closest match with the input time series in each group. The counts are then used to train a ridge regression classifier (Dempster et al., 2023). Here, we use the variant of HYDRA presented in Dempster et al. (2024b), which integrates the HYDRA transform into the process of fitting the ridge regression model, and all computation is performed on GPU.

**Quant** involves recursively dividing the input time series in half, and computing the quantiles for each of the resulting intervals (subseries) (Dempster et al., 2024a). The computed quantiles are used to train an extremely randomised trees classifier. QUANT acts on the original input time series, the first and second derivatives, and the Fourier transform. Here, we use the variant of QUANT presented in Dempster et al. (2024b), which uses pasting to 'spread' the extremely randomised trees over the dataset.

**Extremely Randomised Trees** ('ET') is a well-established classifier, using an ensemble of decision trees where a random subset of features and split points is considered at each node, with the feature/split chosen which minimises log loss (Geurts et al., 2006). Here, we use the same setup as for QUANT, but remove the QUANT transform, so that ET is training directly on the 'raw' time series data (rather than the QUANT features). ET serves as a 'naïve' baseline reference point for the other models.

The four deep learning models are trained using the Adam optimiser (Kingma and Ba, 2015) and a batch size of 256 for a maximum of 100 epochs. The one exception is HInceptionTime with the AudioMNIST dataset, which used a batch size of 64 to enable it to fit in the GPU memory. For all datasets, we implement early stopping and select the best epoch found as the final model, using a validation set obtained by randomly selecting 10% of the training dataset. Training time on each fold is limited to approximately 24 hours or one epoch, whichever is longer.

| mean 0–1 loss | Quant 0.1874 | ConvTran 0.2009 | HInception 0.2020 | Hydra 0.2198 | TempCNN 0.2605 | ET 0.2766 | FCN 0.2940 |
|---|---|---|---|---|---|---|---|
| **Quant** 0.1874 | difference
win / draw / loss
p value | 0.0134
13 / 0 / 15
0.9375 | 0.0145
12 / 0 / 16
0.6295 | 0.0324
20 / 0 / 8
0.0774 | **0.0731**
**19 / 0 / 9**
**0.0014** | **0.0892**
**25 / 0 / 3**
**≤ 1e-04** | **0.1065**
**23 / 0 / 5**
**≤ 1e-04** |
| **ConvTran** 0.2009 | -0.0134
15 / 0 / 13
0.9375 | - | 0.0011
15 / 0 / 13
0.6947 | **0.0189**
**20 / 0 / 8**
**0.0118** | **0.0596**
**21 / 0 / 7**
**0.0013** | **0.0758**
**19 / 0 / 9**
**0.0281** | **0.0931**
**25 / 0 / 3**
**≤ 1e-04** |
| **HInception** 0.2020 | -0.0145
16 / 0 / 12
0.6295 | -0.0011
13 / 0 / 15
0.6947 | - | **0.0178**
**22 / 0 / 6**
**0.0027** | **0.0585**
**23 / 2 / 3**
**0.0006** | **0.0747**
**21 / 0 / 7**
**0.0337** | **0.0920**
**26 / 1 / 1**
**≤ 1e-04** |
| **Hydra** 0.2198 | -0.0324
8 / 0 / 20
0.0774 | **-0.0189**
**8 / 0 / 20**
**0.0118** | **-0.0178**
**6 / 0 / 22**
**0.0027** | - | 0.0407
12 / 0 / 16
1.0000 | 0.0568
13 / 0 / 15
0.3386 | 0.0742
15 / 0 / 13
0.2842 |
| **TempCNN** 0.2605 | **-0.0731**
**9 / 0 / 19**
**0.0014** | **-0.0596**
**7 / 0 / 21**
**0.0013** | **-0.0585**
**3 / 2 / 23**
**0.0006** | -0.0407
16 / 0 / 12
1.0000 | - | 0.0161
18 / 0 / 10
0.4117 | 0.0335
17 / 1 / 10
0.0515 |
| **ET** 0.2766 | **-0.0892**
**3 / 0 / 25**
**≤ 1e-04** | **-0.0758**
**9 / 0 / 19**
**0.0281** | **-0.0747**
**7 / 0 / 21**
**0.0337** | -0.0568
15 / 0 / 13
0.3386 | -0.0161
10 / 0 / 18
0.4117 | - | 0.0173
15 / 0 / 13
0.5369 |
| **FCN** 0.2940 | **-0.1065**
**5 / 0 / 23**
**≤ 1e-04** | **-0.0931**
**3 / 0 / 25**
**≤ 1e-04** | **-0.0920**
**1 / 1 / 26**
**≤ 1e-04** | -0.0742
13 / 0 / 15
0.2842 | -0.0335
10 / 1 / 17
0.0515 | -0.0173
13 / 0 / 15
0.5369 | - |

−0.1    0    0.1

difference

Figure 12: Multi-comparison matrix showing mean 0–1 loss and pairwise differences.

We provide results for 0–1 loss, log loss, weighted F1 score, balanced accuracy, and training time. Each method is evaluated on each dataset using 5-fold cross-validation, using predefined cross-validation folds. (Note that both QUANT and ET are unable to train on one of the folds of the WISDM dataset, due a limitation of the ET implementation where there is a single example of a given class.) These results serve as an initial survey on the relative performance of different methods on the MONSTER datasets, to serve as a reference point for future work on large time series classification tasks.

### 4.2 Summary

The multi-comparison matrix (MCM) in Figure 12 shows mean 0–1 loss as well as pairwise differences and win/draw/loss for the baseline methods over all 28 MONSTER datasets (see Ismail-Fawaz et al., 2023a).

Figure 12 shows that QUANT achieves the lowest overall mean 0–1 loss, slightly lower than that of ConvTran, although both ConvTran and HInceptionTime have lower 0–1 loss on more datasets (15 vs 13 and 16 vs 12 respectively). HYDRA has higher overall mean 0–1 loss than HInceptionTime, but lower than TempCNN, ET, or FCN. TempCNN, ET, and FCN all have higher average 0–1 loss, due in large part to poor performance on the audio datasets: see Section 4.3.

### 4.3 By Category

Figure 13 shows the 0–1 loss for each method on each dataset, organised by category (Audio, Count, EEG, HAR, Satellite, and Other). Each point represents a single dataset. The

horizontal bars represent mean 0–1 loss for each classifier within each category. Figure 13 shows that while for some categories the 0–1 loss for different methods is broadly similar, for other categories there are considerable differences.

In particular, ConvTran, HInceptionTime, Hydra and Quant all achieve relatively low 0–1 loss on the audio datasets, while ET, TempCNN, and especially FCN have much higher 0–1 loss. ET, Quant, and (to a lesser extent) ConvTran and HInception achieve relatively low 0–1 loss on the count datasets. Quant and (to a lesser extent) ET and Hydra achieve relatively low 0–1 loss on the 'other' datasets.

In contrast, mean 0–1 loss for EEG, HAR, and Satellite is broadly similar, with significant spread within the results for each method.

Interestingly, it is only on the audio datasets, and to some extent the HAR datasets, that our naïve baseline, ET, appears to be meaningfully 'worse' than the deep learning or specialised time series classification methods. ET achieves similar results to Quant on a number of datasets, which is not surprising, as the 'raw' time series are similar to a subset of the features used in Quant.

We speculate that the poor 0–1 loss for FCN and TempCNN on the audio datasets in particular may be related to the small receptive field of these models (relative to the relatively long time series in the audio datasets). With a small receptive field, these models are in effect limited to high-frequency features in the data.

The satellite datasets show some interesting extremes. All methods except for Quant and ET performed poorly on the *S2Agri* 10% datasets. In contrast, all methods achieved very low 0–1 loss on *LakeIce* as this dataset has strong temporal and spatial correlations between samples that could not be accounted for when splitting the data into folds.

These differences in the relative performance of different algorithms on different types of learning task lends support to the prospect that benchmarks using larger training sets will promote research into matching the prior assumptions of different learning algorithms to different types of learning task.

Figures 18 and 19 (Appendix A) show weighted F1 score and balanced accuracy for each method organised by category. As for Figure 13, each point represents a dataset, and the horizontal bars represent the mean score for each classifier within each category. Overall, weighted F1 score broadly follows 0–1 loss, although with a greater spread of values. Balanced accuracy shows a greater spread of values again, particularly for satellite image time series, where significant class imbalance appears to result in very low balanced accuracy for a number of datasets, particularly for FCN, HInception, TempCNN, and Hydra. (Interestingly, FCN, HInception, and TempCNN all achieve high weighted F1 scores and balanced accuracy on one of the EEG datasets.)

## 4.4 Computational Efficiency

### 4.4.1 Training Time

Table 2 shows total training time for each of the baseline methods, separated into methods using GPU and methods using CPU. This represents the total training time over all 28 Monster datasets (where the time for each dataset is the average training time across the five cross-validation folds). These training times are intended to provide an approximate, real-world estimate of the training time required for the different methods presented here.

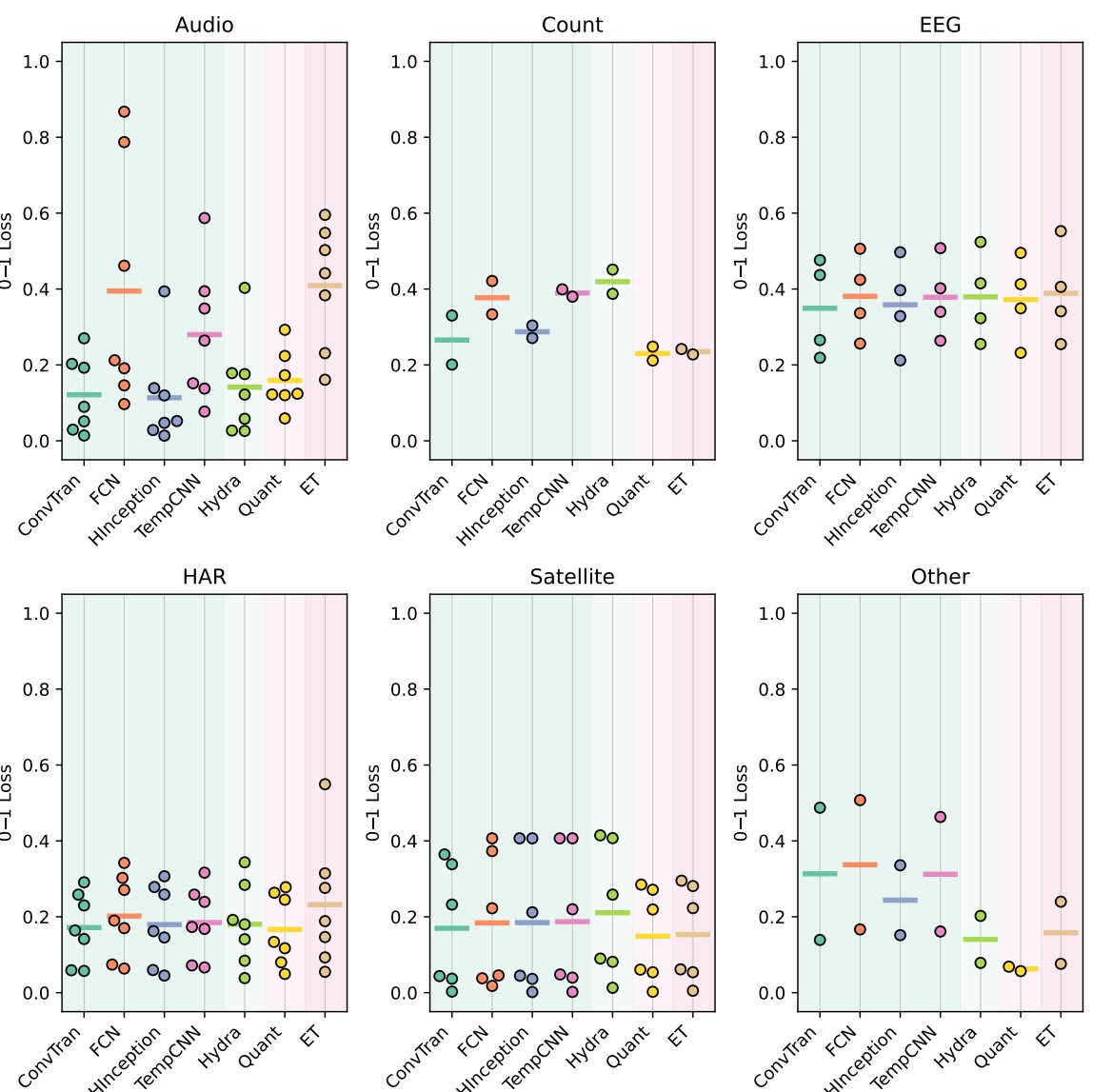

Figure 13: 0–1 loss by category.

Table 2: Total Training Time

| GPU | | | | | CPU | |
|---|---|---|---|---|---|---|
| Hydra | ConvTran | TempCNN | FCN | HInception | ET | Quant |
| 39m 19s | 5d 2h | 2d 7h | 2d 10h | 5d 18h | 11h 6m | 1d |

Table 3: Number of Parameters

| | ConvTran | FCN | HInception | TempCNN | Hydra† | Quant‡ |
|---|---|---|---|---|---|---|
| *min* | 27,039 Traffic | 264,962 CornellWhale | 869,570 CornellWhale | 424,649 Tiselac | 6,144 FordChallenge | 275 CrowdSourced |
| *max* | 486,941 Opportunity | 380,037 Opportunity | 1,420,145 Opportunity | 786,444,426 AudioMNIST | 167,936 Pedestrian | 379,112 Traffic |

† num. parameters in ridge classifier; ‡ median num. leaves per tree

However, we note that as different methods have been trained using a mixture of different GPUs and CPUs, these timings are not directly comparable, and can only provide a rough estimate of their comparative computational efficiency.

The five methods trained using GPUs were each trained using a single GPU, either an Ampere A100 SMX4 with 80GB RAM, or an Ampere A40 with 48GB RAM. Table 2 shows that among these methods, HYDRA is by far the fastest, taking less than 40 minutes to train over all 28 MONSTER datasets, more than $80\times$ faster than the next-fastest GPU method (TempCNN). HInceptionTime is the least efficient method, requiring more than five days of training time, corresponding to almost one month total training time across all five cross-validation folds. (We note that there is a variant of HInceptionTime, LITETime, with significantly fewer parameters which requires less than half of the training time of HInceptionTime: Ismail-Fawaz et al. (2023b).)

Although not directly comparable to methods using GPU, QUANT requires approximately 24 hours of training time (using 4 CPU cores). ET requires less than half of this (approx. 11 hours), due to the smaller number of features used to train the classifier.

### 4.4.2 PARAMETER COUNTS

Table 3 shows total number of parameters for each of the baseline methods. For each method the table shows the minimum and maximum number of parameters and the corresponding dataset. The number of parameters for both FCN and HInceptionTime is reasonably stable, with the largest model 1.4 and 1.6 times that of the smallest model, respectively. However, the number of parameters in the TempCNN models vary greatly, with the largest model being over 1,800 times the size of the smallest one. While the total number of parameters for all the deep learning methods is dependent on the number of classes and channels, the FCN and HInceptionTime architectures both include a global average pooling layer, so the parameter count is independent of the length of the time series. However, TempCNN does not use global pooling and so its parameter count is highly dependent on the length of the time series. For HYDRA, we have used the number of parameters for the ridge classifier, and for QUANT we have used the median number of leaf nodes, although these are not directly comparable to the number of trainable parameters in the deep learning models.

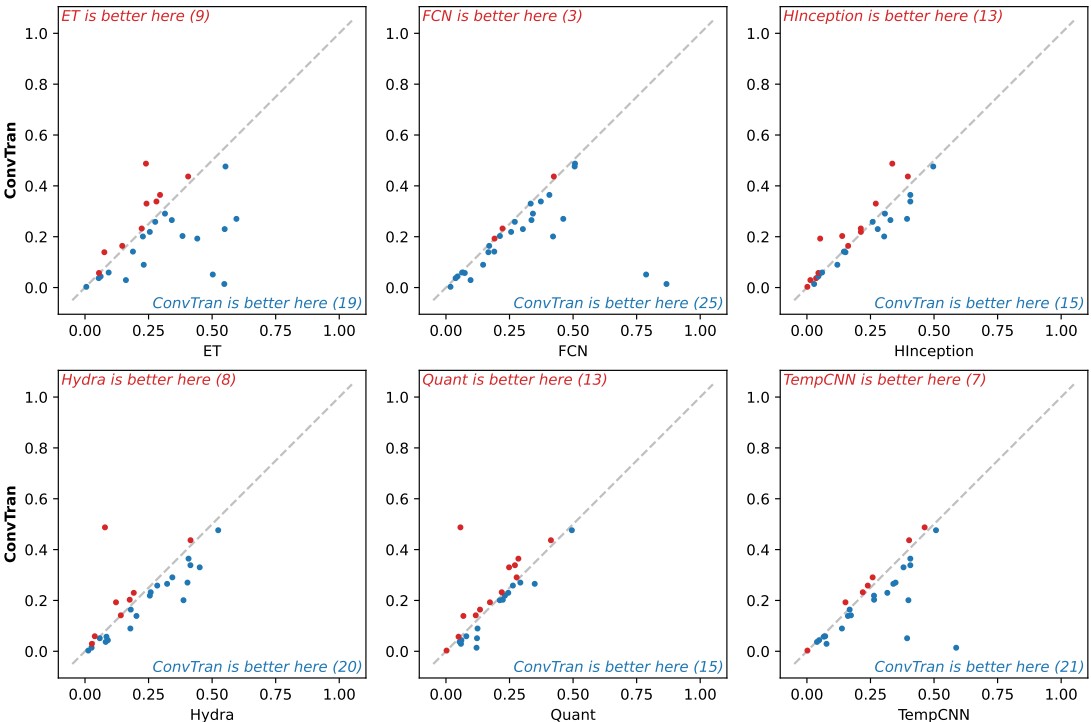

Figure 14: Pairwise **0–1 loss** for ConvTran.

## 4.5 Pairwise Comparisons

Figures 14, 15, and 16 show the pairwise 0–1 loss, log loss, and training time for ConvTran versus each of the other baseline methods. (Full pairwise results for all methods and metrics are provided in the Appendix.) Figure 14 shows that ConvTran achieves broadly similar 0–1 loss on most datasets compared to Quant, HInceptionTime, and Hydra (both Hydra and Quant achieve significantly lower 0–1 loss than ConvTran on one dataset).

While ConvTran achieves similar 0–1 loss to FCN and TempCNN on most datasets, ConvTran achieves considerably lower 0–1 loss on a small number of datasets. As noted above, these include the audio datasets, where FCN and TempCNN appear to struggle relative to the other methods.

Figure 15 shows a slightly different picture in terms of log loss. ConvTran is fairly evenly matched to Quant (and ET) in terms of the number of datasets on which each method achieves lower log loss, although there is a considerable spread in values (i.e., they are not closely correlated). ConvTran achieves lower log loss on more datasets compared to Hydra, although Hydra does achieve lower log loss on 10 datasets, which is somewhat surprising, as Hydra takes no account of log loss in training. ConvTran achieves lower log loss than FCN, HInceptionTime, and TempCNN on most datasets.

Figure 16 shows that ConvTran is significantly faster than HInceptionTime, but slower than FCN or TempCNN, on most datasets. ConvTran is marginally faster than Quant and ET on a number of datasets, although the timings are not directly comparable (given that

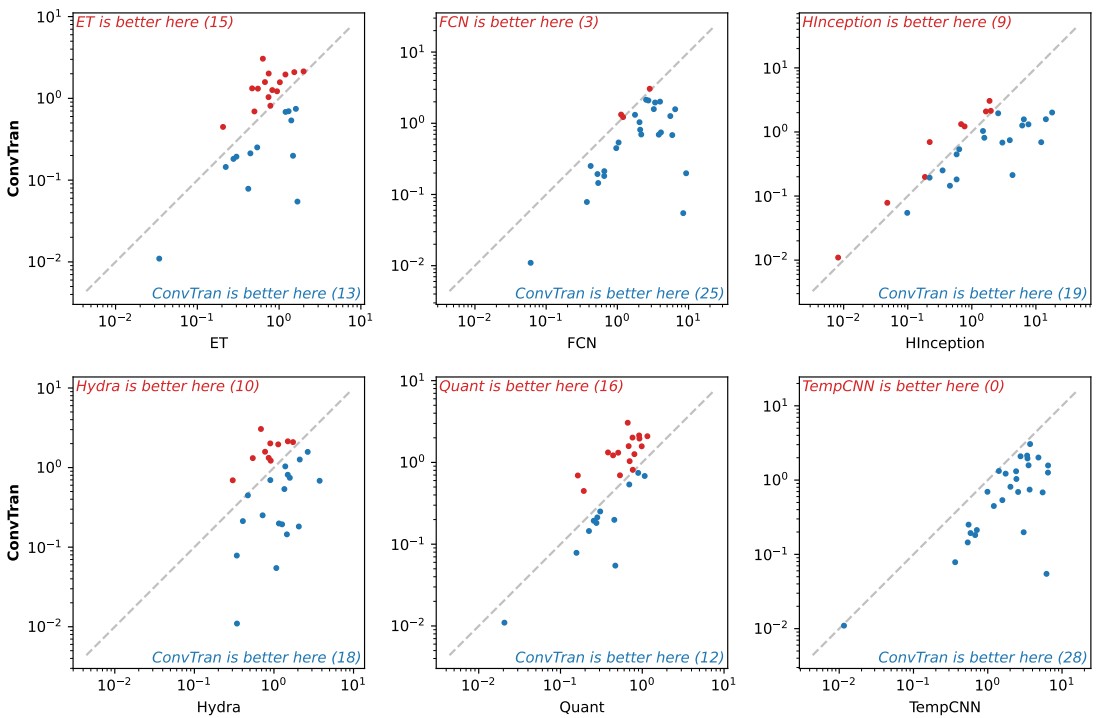

Figure 15: Pairwise **log-loss** for ConvTran.

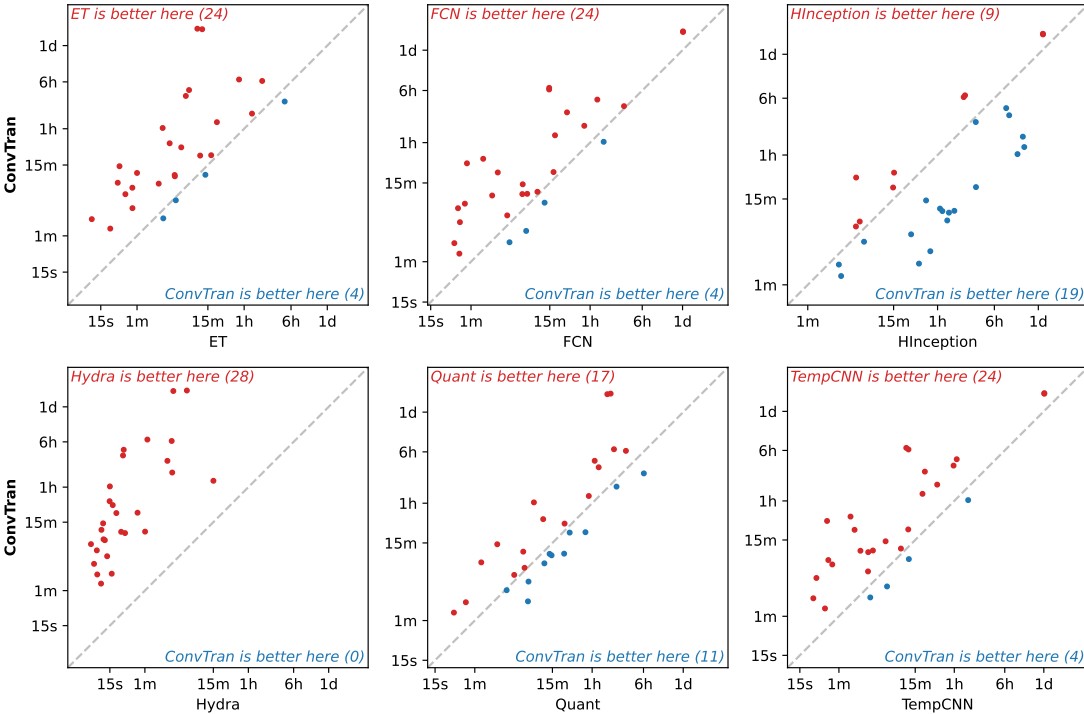

Figure 16: Pairwise **training time** for ConvTran.

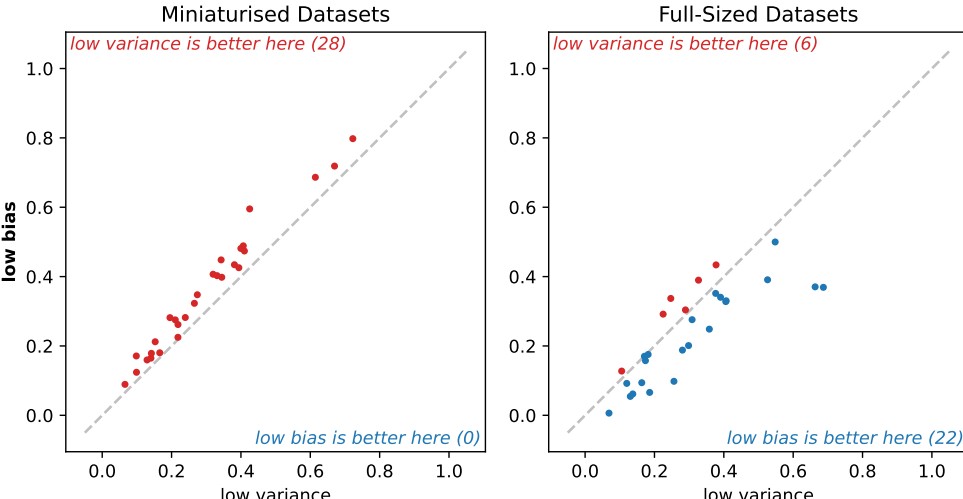

Figure 17: Pairwise **0–1 loss** for a low variance configuration of QUANT versus a lower bias configuration of QUANT on: miniaturised versions of the MONSTER datasets (left) and the full-sized MONSTER datasets (right).

ConvTran uses GPU whereas QUANT is limited to CPU). HYDRA is faster than ConvTran on all datasets (reflecting the overall differences in training time shown in Table 2).

## 4.6 Training Set Size and the Bias–Variance Trade-Off

We have sought to use the bias–variance trade-off to motivate the value of time series classification benchmarks using much larger training sets than those in the UCR and UEA archives. While learning curves, such as Figure 1, above, can give credence to this argument, it would be desirable to provide more substantive evidence. Unfortunately, any attempt to do so is complicated by the issue that while algorithms can be inherently low variance, their bias component of error is a function of the degree of fit of their prior assumptions to the requirements of the learning task (i.e., inductive bias). It is further complicated by a little recognised issue, that learning tasks often involve an element of concept shift, where the the data distribution in the training data does not exactly match the distribution in the test data. As a result, a low bias algorithm that perfectly learns the classification function that gave rise to the training data may have higher error on the test data than a higher bias algorithm that has a less perfect fit.

To address the first of these issues we created a learning algorithm pair that share the same overarching prior assumptions about the learning task, but one relaxes those assumptions relative to the other resulting in a higher-variance, lower-bias variant. To this end we developed two different configurations of QUANT. As for Figure 1, above, the low variance configuration of QUANT uses a maximum tree depth of 4 and 128 trees, while the low bias configuration of QUANT uses unlimited tree depth and 4 trees. Figure 17 shows pairwise 0–1 loss for the low variance configuration of QUANT versus the low bias configuration of QUANT on both miniaturised versions of the MONSTER datasets (left), and

the full MONSTER datasets (right). The miniaturised benchmark was created by taking a stratified sample of 200 training examples for each cross-validation fold, to approximately match the median training set sizes of 217 examples and 255 examples in the UCR and UEA archives respectively.

Figure 17 shows that the low variance model achieves lower 0–1 loss on all 28 small datasets. In contrast, the low bias model achieves lower 0–1 loss (considerably lower in many cases) on 22 of the 28 full-sized datasets. All 6 of the datasets where the low variance model achieves lower 0–1 loss are HAR or EEG datasets with very challenging subject-wise cross-validation folds which conceivably introduce substantial concept shift between the training and test sets. Figure 17 very clearly demonstrates the imperative of reducing variance for smaller quantities of training data, and reducing bias for larger quantities of training data. It further suggests that large training set benchmarks may motivate greater attention for the issue of concept shift in time series classification.

## 5 Conclusion

We present MONSTER, a new benchmark collection of large datasets for time series classification. The field of time series classification has become focused on smaller datasets. This has resulted in state-of-the-art methods being optimised for low average 0–1 loss over a large number of small datasets, has insulated the field from engaging with the challenges of learning from large quantities of data, and has artificially disadvantaged low-bias methods such as deep neural network models in benchmarking comparisons.

We hope that MONSTER encourages the field to engage with the challenges related to learning from large quantities of time series data. We hope that MONSTER will help better reflect the broader task of time series classification and improve relevance for real-world time series classification problems. We believe there is enormous potential for new research based on much larger time series datasets; research that addresses the engineering challenges of learning from massive data; research that matches the prior assumptions of different learning algorithms to the requirements of different learning tasks; and research that addresses the issue of concept shift between training and test data.

## Broader Impact Statement

We present a new benchmark of 28 large datasets for time series classification. This could potentially have a large impact on the field, as these datasets are significantly larger than those currently used for benchmarking and evaluation. This should allow for training lower-bias, more complex models, with greater relevance and more direct applicability to large-scale, real-world time series classification problems. On the other hand, learning from larger quantities of data requires proportionally more computational time and resources. As such, it is important to always keep in mind the balance between computational expense and real-world relevance. There are also potential risks associated with the misuse of improved methods for time series classification in monitoring and surveillance, although we do not feel that there is any significant direct risk associated with this work.

## Acknowledgments and Disclosure of Funding

This work was supported by an Australian Government Research Training Program Scholarship, and the Australian Research Council under award DP240100048. The authors would like to thank, in particular, Professor Eamonn Keogh, Professor Tony Bagnall, and all the people who have contributed to the UCR and UEA time series classification archives. The authors also thank Raphael Fischer for trialling our methods and datasets and providing invaluable feedback.

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

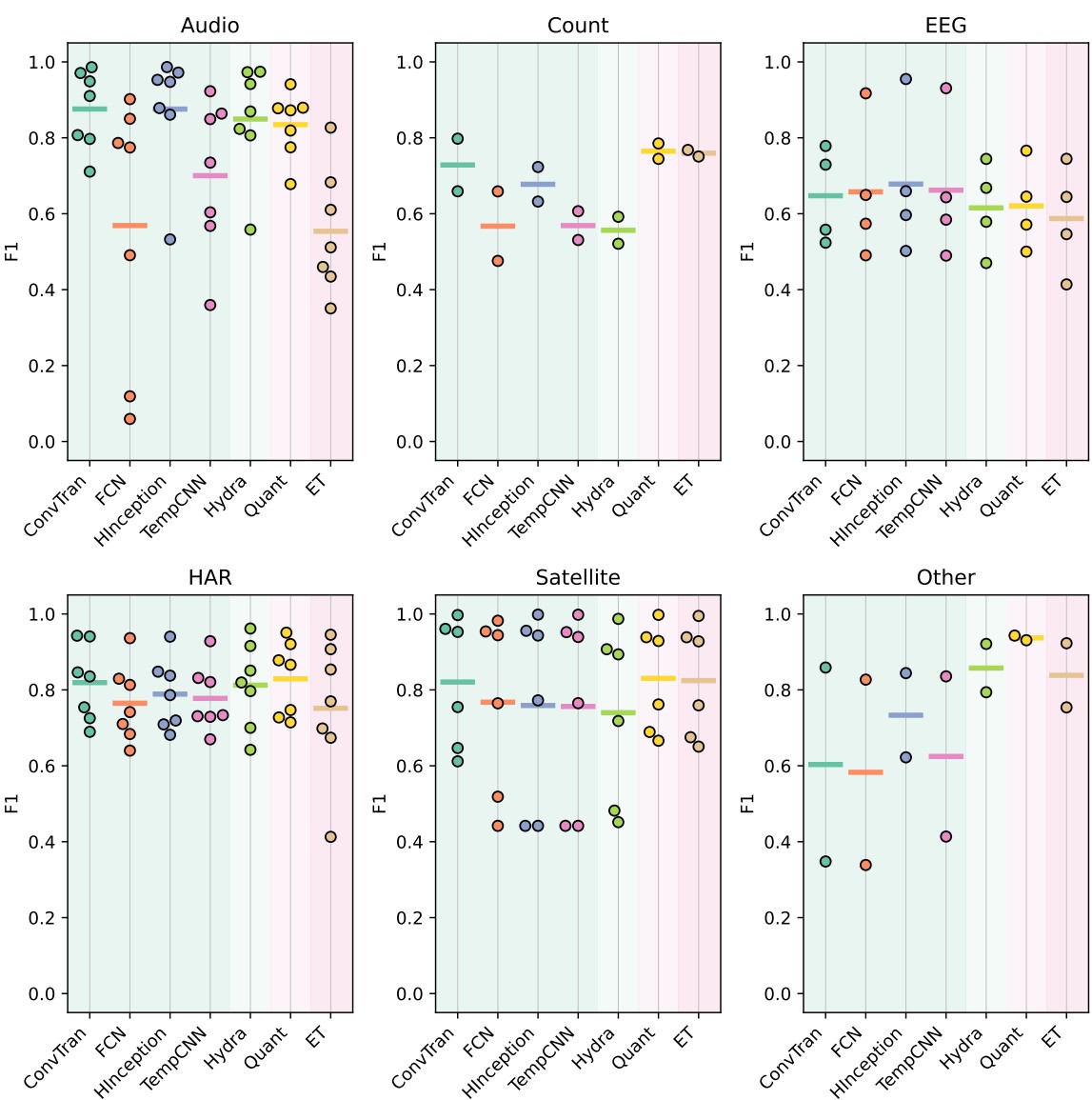

# Appendix A. Additional Results

## A.1 Weighted F1 Score

Figure 18: Weighted F1 score by category.

## A.2 Balanced Accuracy

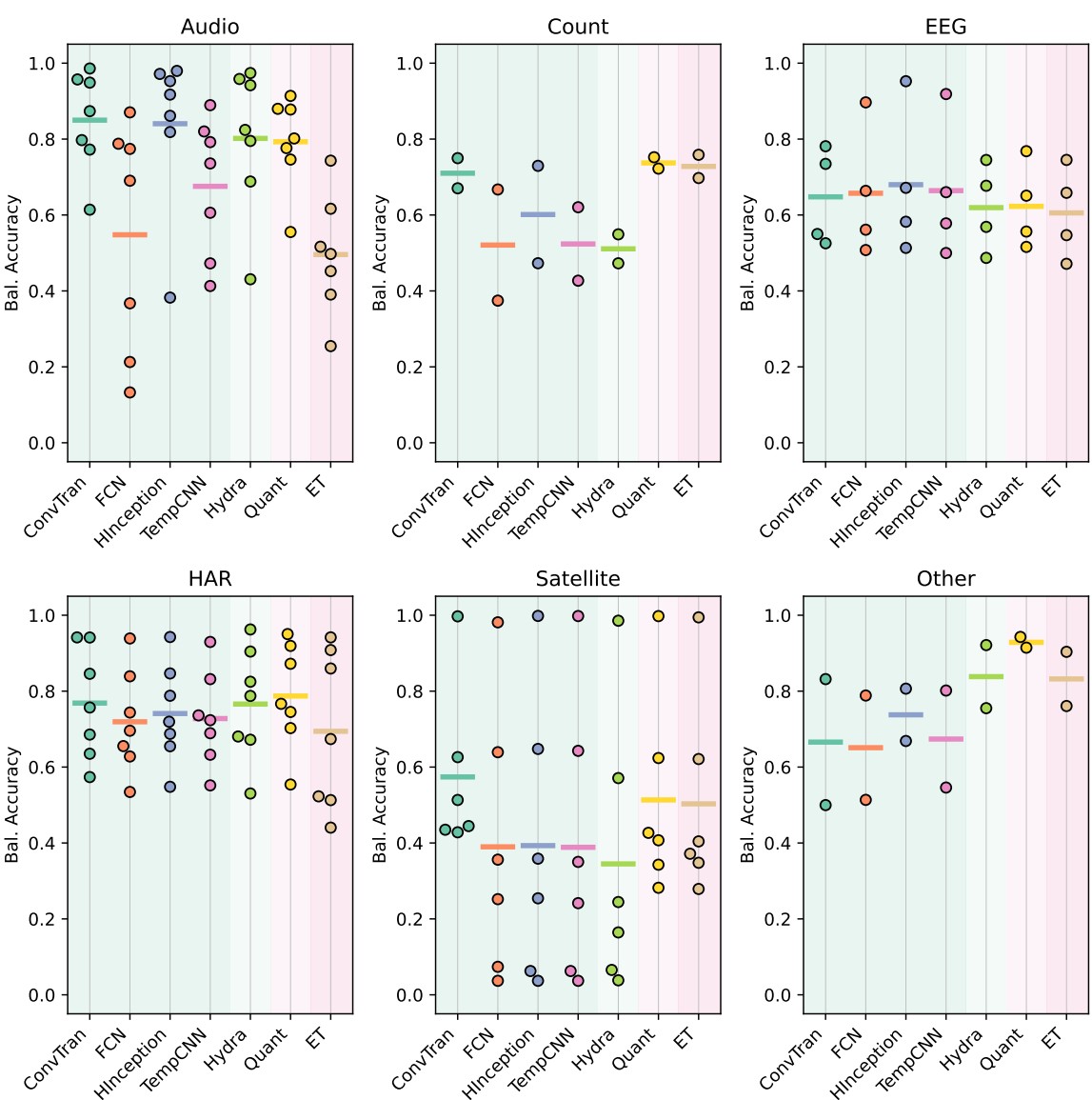

Figure 19: Balanced accuracy by category.

## A.3  0–1 loss

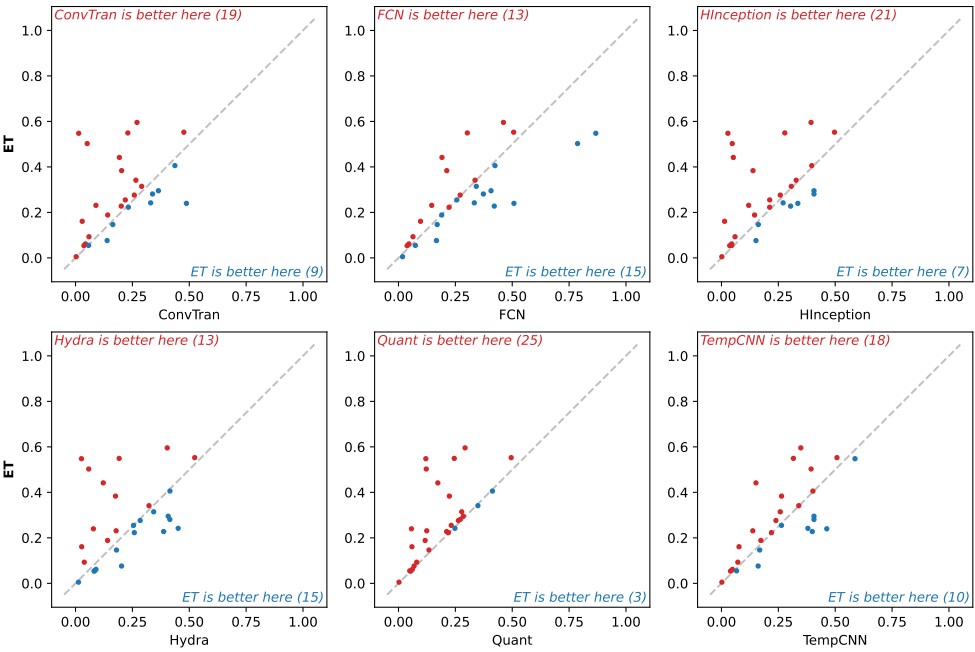

Figure 20: Pairwise results (0–1 loss) for ET.

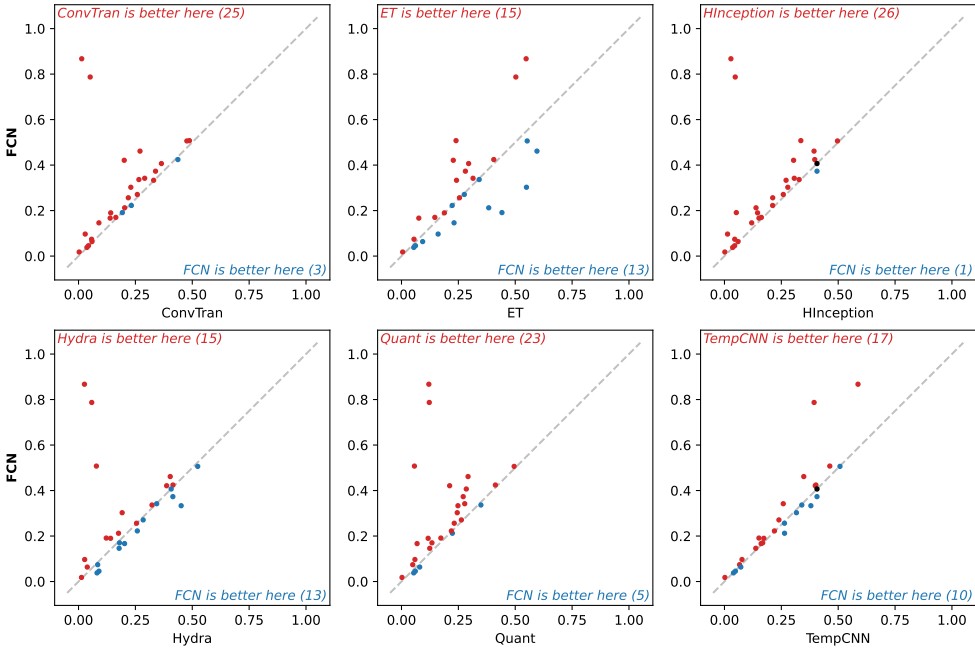

Figure 21: Pairwise results (0–1 loss) for FCN.

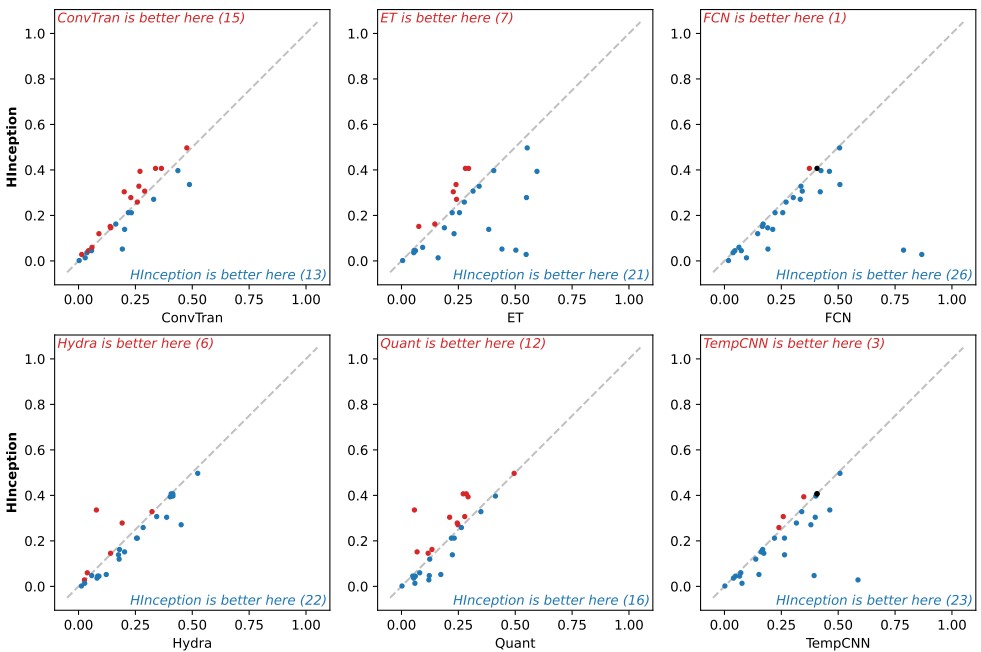

Figure 22: Pairwise results (0–1 loss) for HInceptionTime.

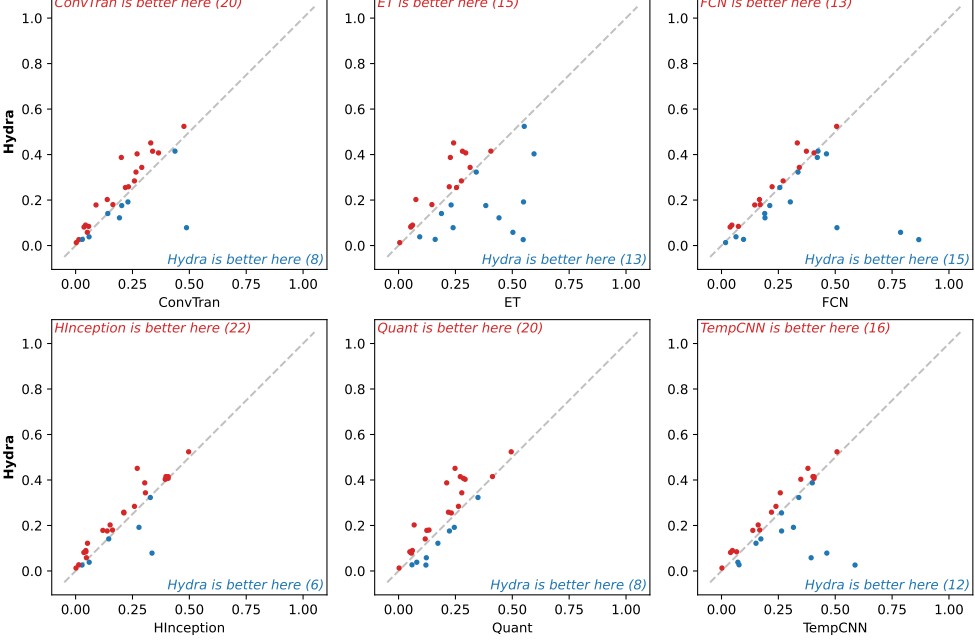

Figure 23: Pairwise results (0–1 loss) for HYDRA.

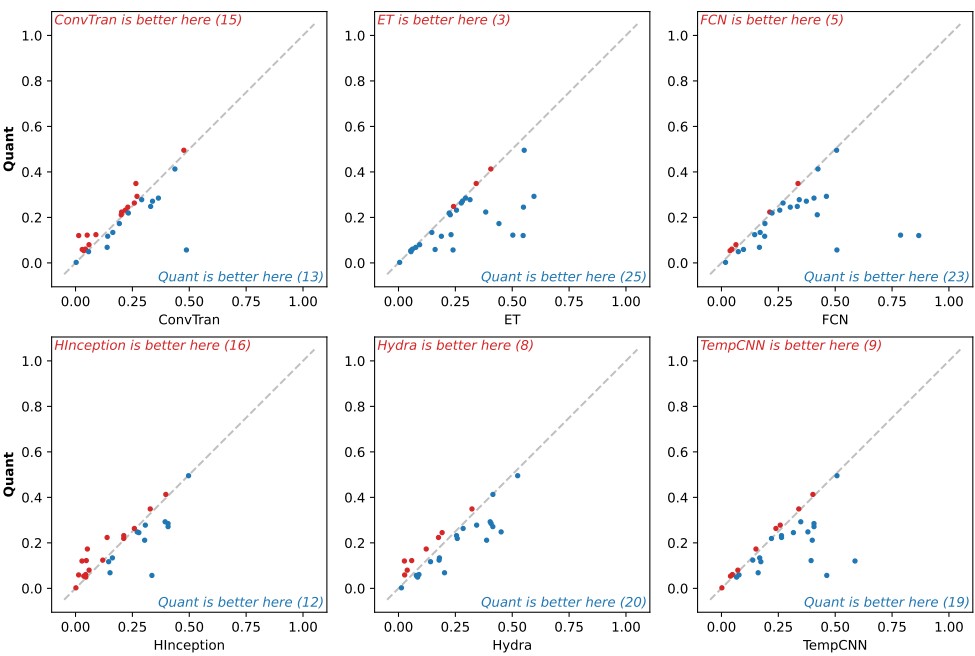

Figure 24: Pairwise results (0–1 loss) for Quant.

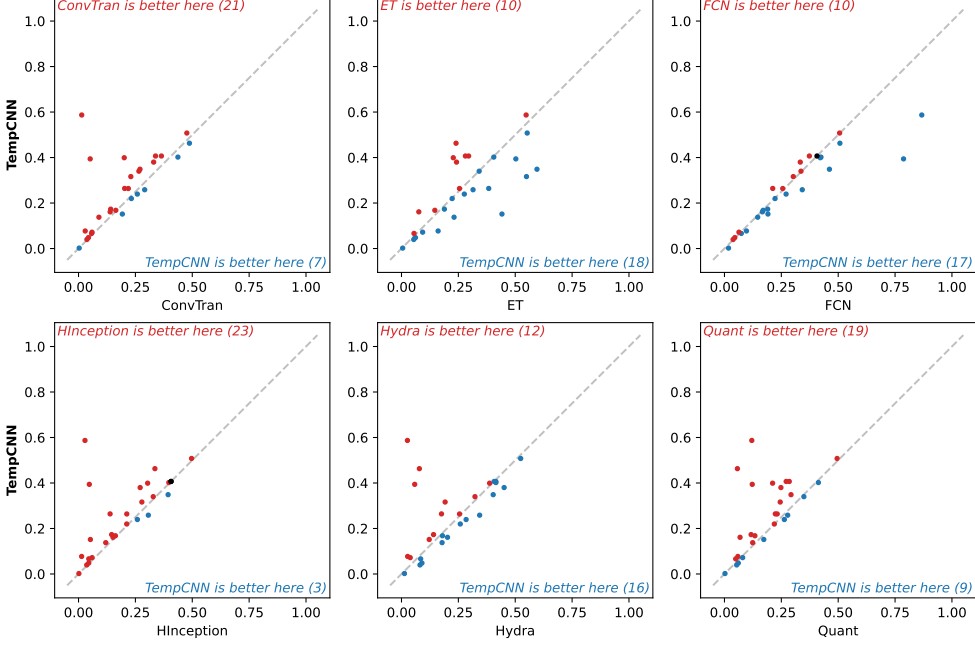

Figure 25: Pairwise results (0–1 loss) for TempCNN.

## A.4 Log Loss

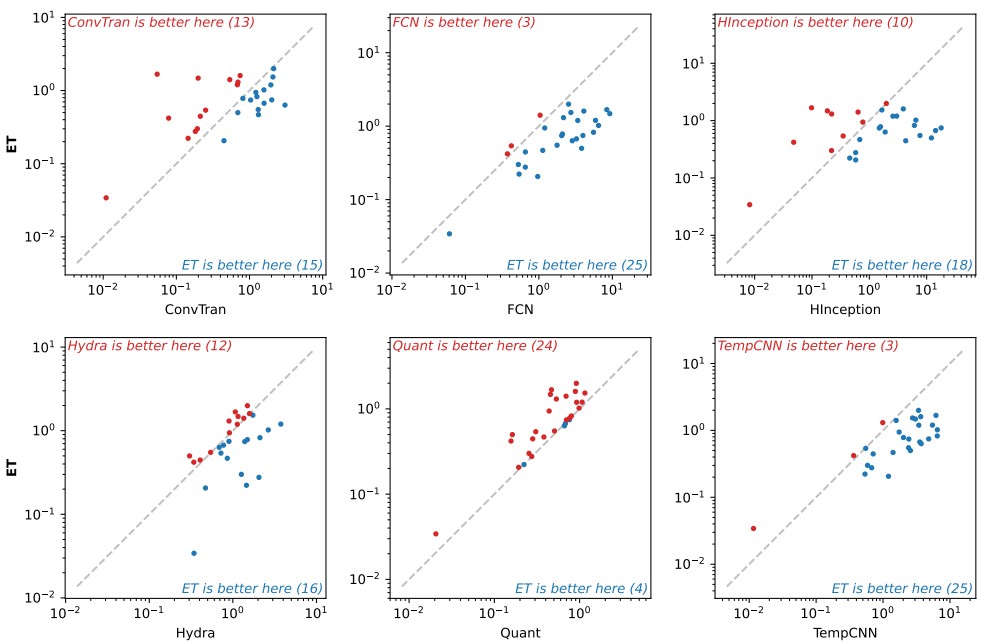

Figure 26: Pairwise results (log-loss) for ET.

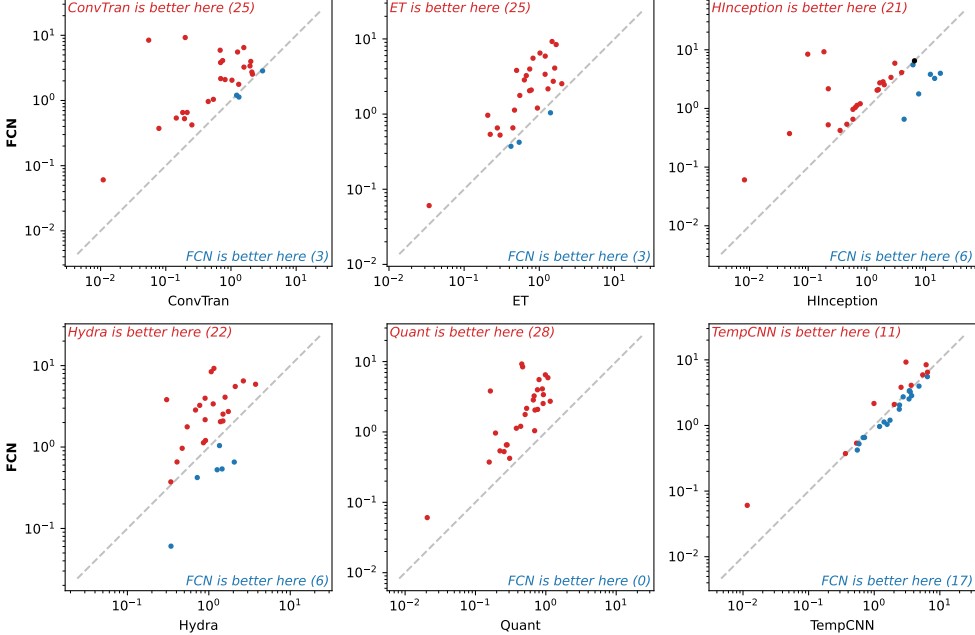

Figure 27: Pairwise results (log-loss) for FCN.

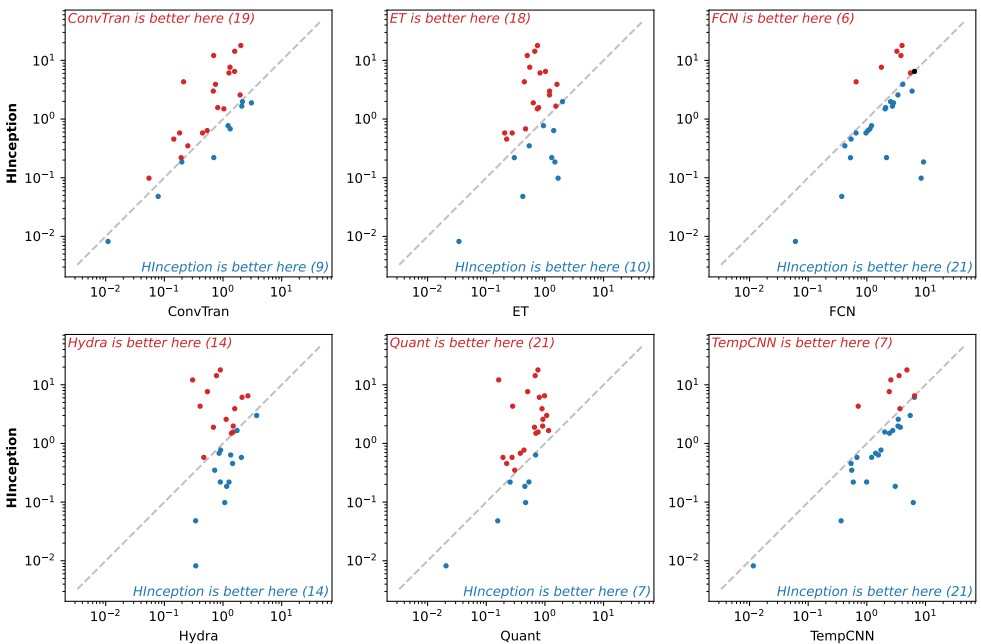

Figure 28: Pairwise results (log-loss) for HInceptionTime.

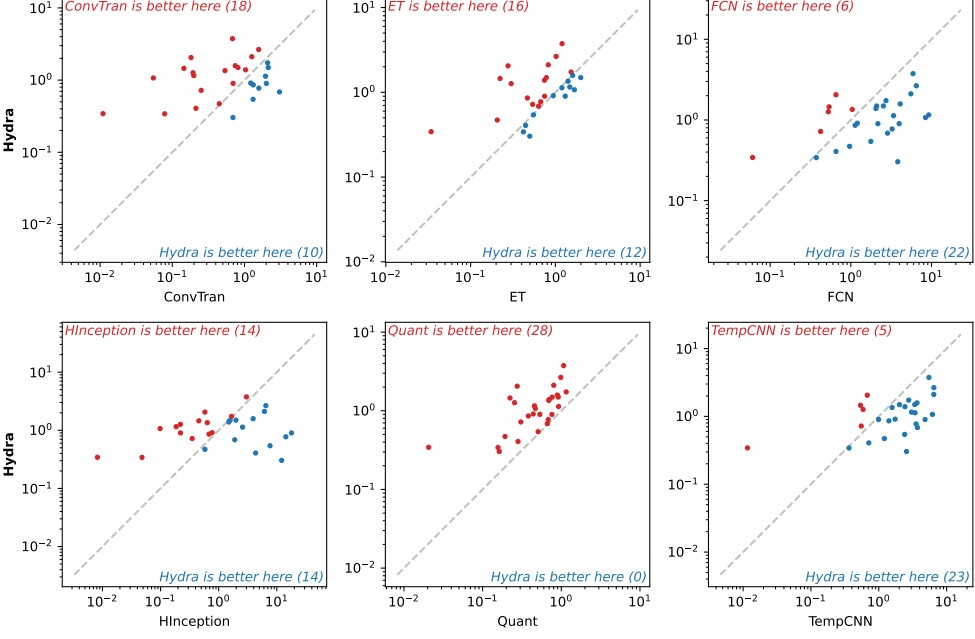

Figure 29: Pairwise results (log-loss) for HYDRA.

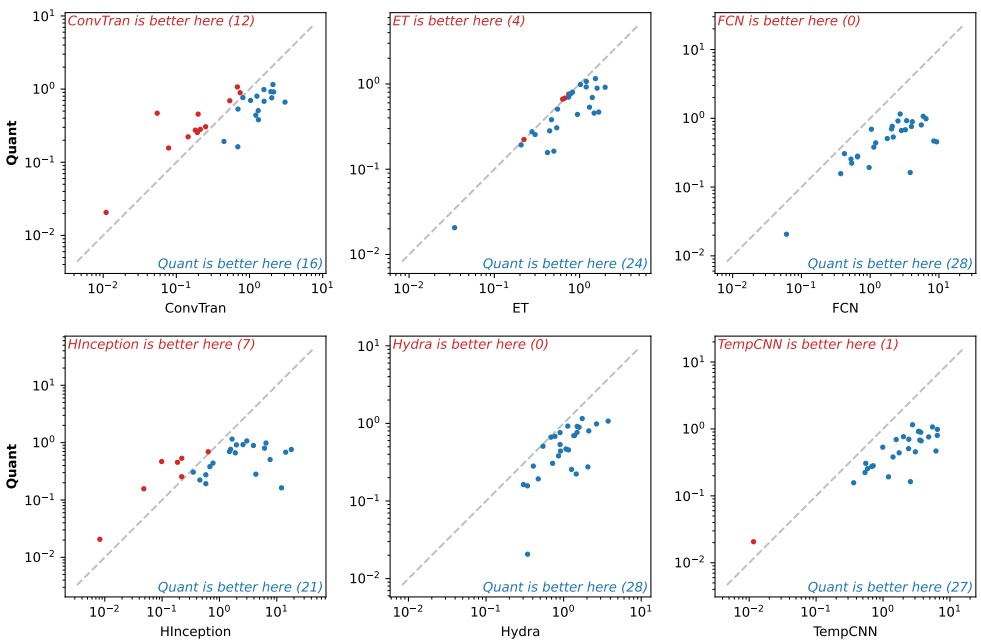

Figure 30: Pairwise results (log-loss) for QUANT.

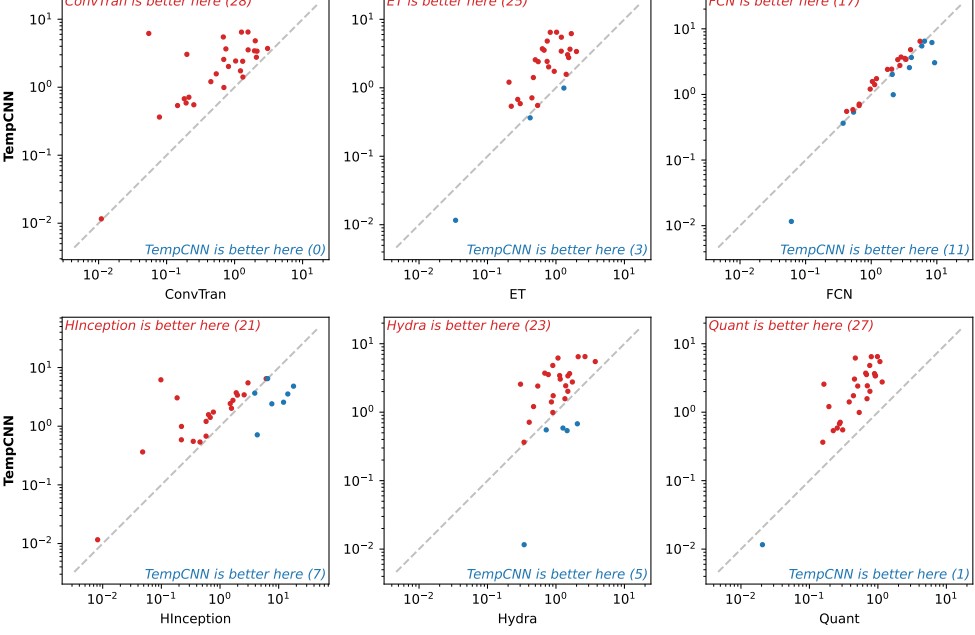

Figure 31: Pairwise results (log-loss) for TempCNN.

## A.5 Training Time

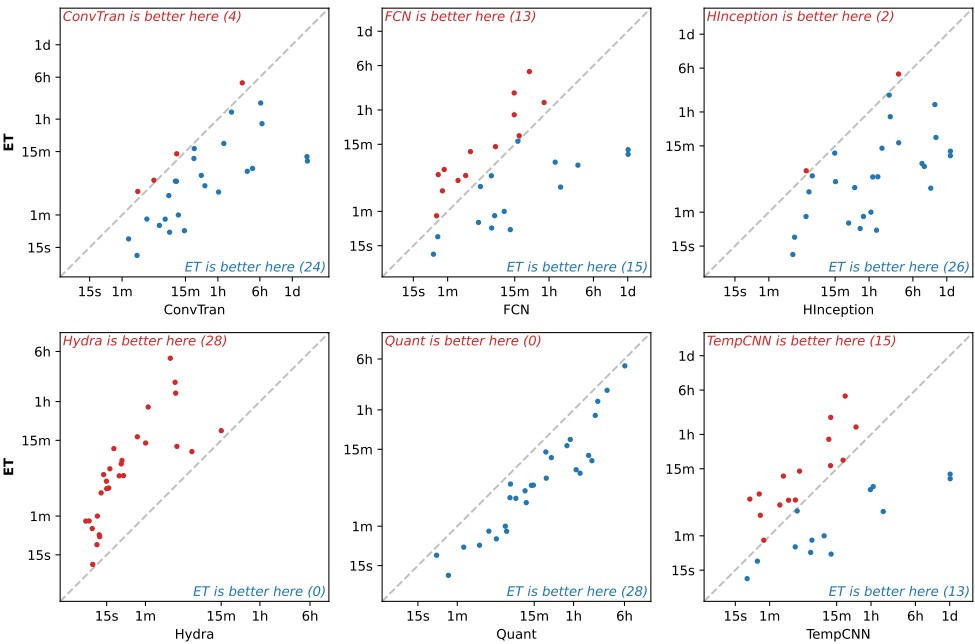

Figure 32: Pairwise results (training time) for ET.

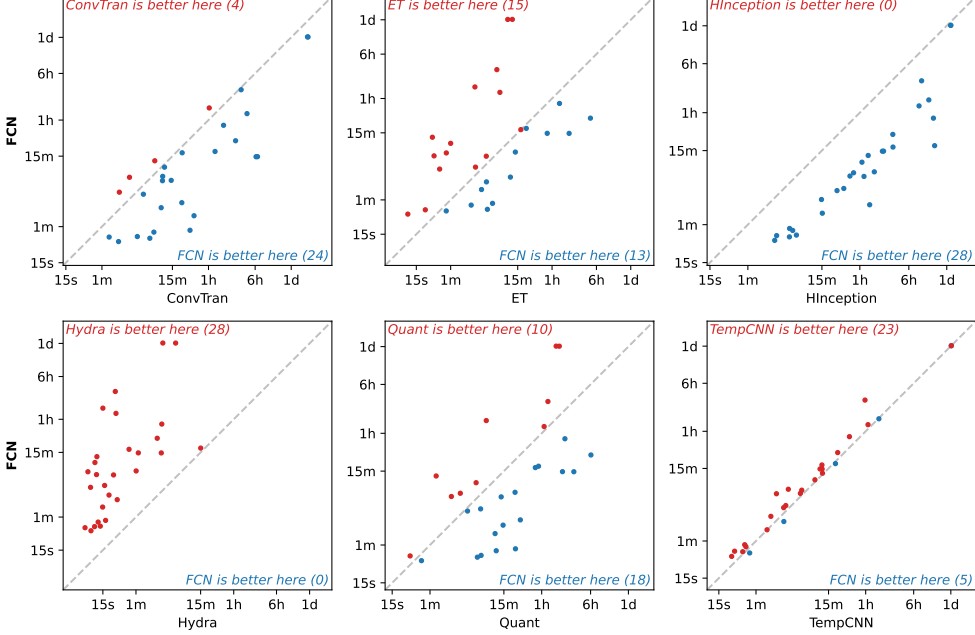

Figure 33: Pairwise results (training time) for FCN.

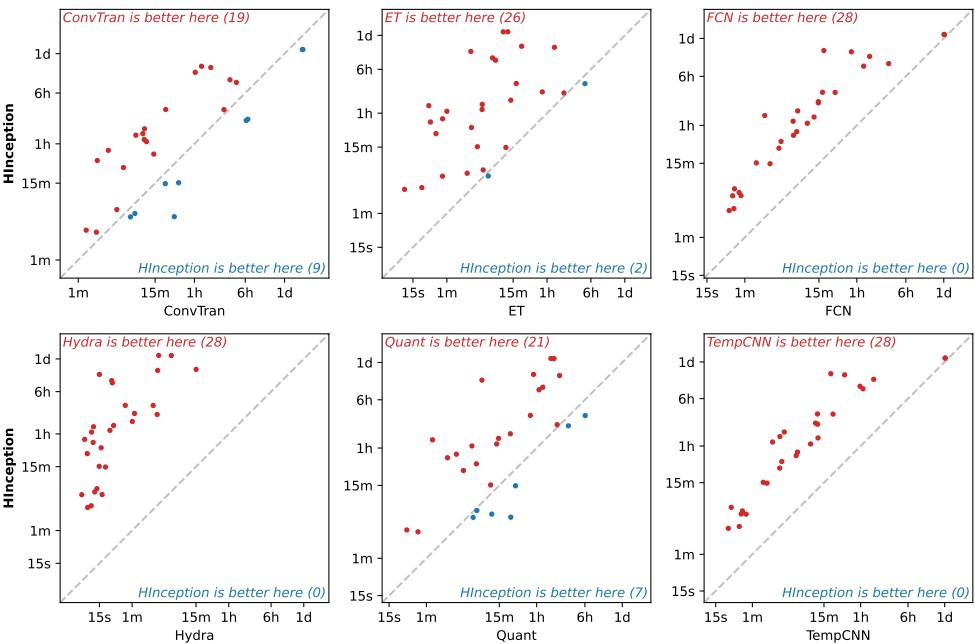

Figure 34: Pairwise results (training time) for HInceptionTime.

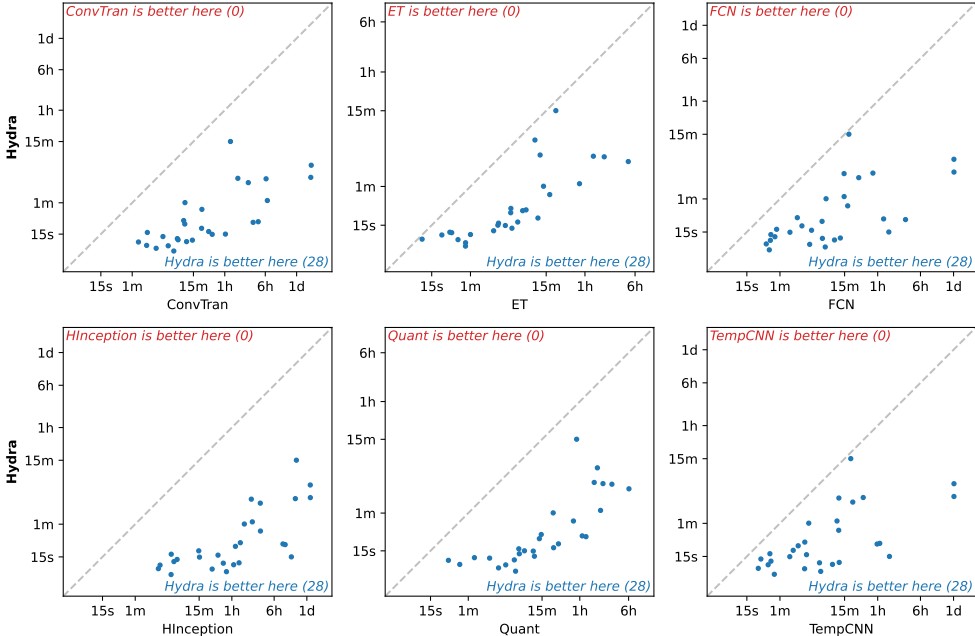

Figure 35: Pairwise results (training time) for HYDRA.

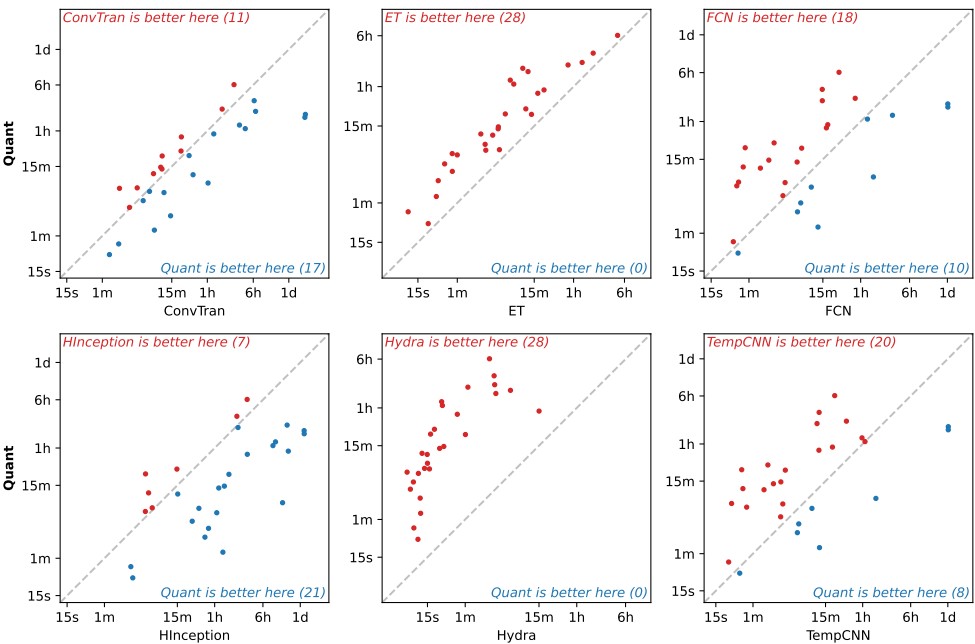

Figure 36: Pairwise results (training time) for QUANT.

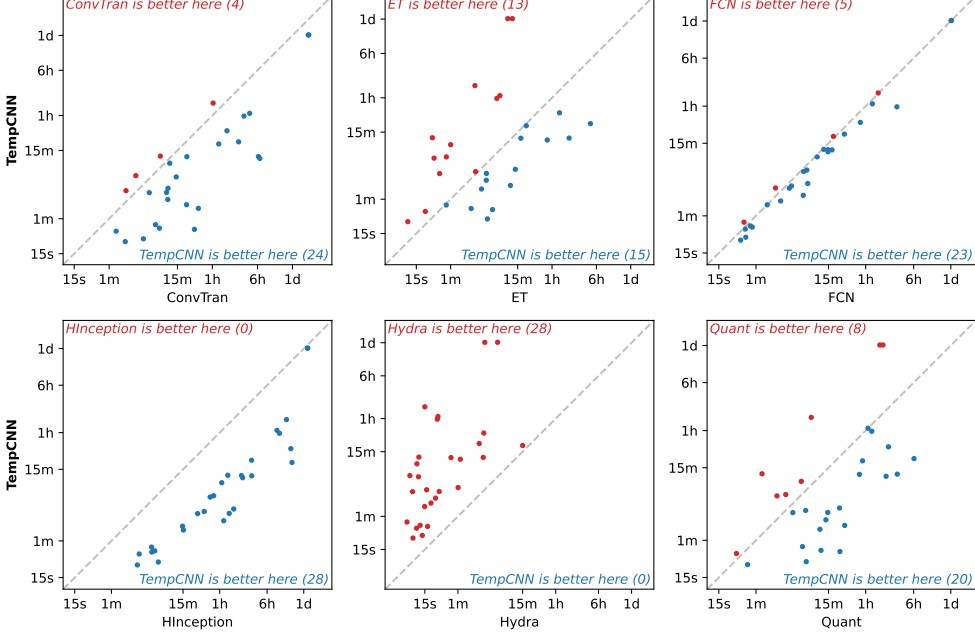

Figure 37: Pairwise results (training time) for TempCNN.

## Appendix B. Hosting

We consider this as an initial release, and we aim to continue to add datasets to the benchmark. We will endeavour to promptly address any issues that arise with the datasets and provide updated versions of the datasets where relevant.

We intend to host the MONSTER datasets via HuggingFace indefinitely: `https://huggingface.co/monster-monash`. However, we also maintain master copies of each of the datasets and, in case it becomes necessary to provide an alternative hosting channel, we will make the datasets available via another platform. We intend to continue to add additional datasets to the MONSTER benchmark over time. We will create new versions of the datasets to reflect any changes or corrections, while keeping older and original versions of the datasets available.

