# OpenReview forum: "MONSTER: Monash Scalable Time Series Evaluation Repository"
_DMLR — Accepted by DMLR_

### Review · Reviewer_51oC · 2025-04-01

**Recommendation:** 4
**Confidence:** 2

**Summary Of Contributions:**

Paper presents a new data repository, collecting together different large-scale time-series classification datasets for benchmarking and machine learning (ML) model development. The work is motivated by the limitation of current, pretty small, time-series dataset repositories, utilised to compare and benchmark new ML models, and are typically only concentrating on minimising 1-0 loss with low-variance (high-bias) methods. A collection of 29 different datasets in 6 different categories are introduced, where the characteristics of each datasets are described. Example benchmark comparison study with some of the state-of-the-art models, including deep learning models and few more traditional ML models, are presented by looking at 1-0 loss, log-loss, and training times with some interesting results.

**Strengths:**

Paper proposes important contribution to time-series classification benchmarking. Although many of the datasets have been available from different sources before, this work brings them to a common repository for easy access that could support time-series model development and comparison, providing valuable information and tools for the ML community. Paper is clearly written and structured.

**Audience:**

Yes

**Broader Impact Concerns:**

Broader Impact Statement is present in the paper. I don't find any concerns on that.

**Claims And Evidence:**

Most of the paper claims are supported; 1) introducing the dataset collection with novel large-scale (e.g., large number of example, longer time-series) with proper web links and access details, 2) initial benchmark, showing the some aspects that new benchmarking is needed, chosen method could depend on particular application, and (over-parameterised) deep learning models are not always the best ones in these kind of problems.

What is a bit limited, and described above, is the link to previous small-scale datasets and how the findings of the proposed benchmark compares to these datasets, considering the bias-variance trade-off. Also, the more versatile performance metrics could support the findings even better.

**Datasets And Benchmarks:**

Datasets are described sufficiently most of the parts (see some suggestions above), including formats and details of each datasets, web links to repository, information about licensing etc., and how to utilise the datasets. Details of the example benchmarking is also given and link to code repository to support reproducibility.

**Extended Submissions:**

As far as I know, this is a new submission.

**Limitations:**

List of weaknesses and limitations
- Missing some dataset details: temporal and spatial resolution (in real units), sensors names/types
- Missing detailed description of temporal characteristics of different dataset categories and input data types
- Weak analysis of bias-variance trade-off challenge (in the experiments) in relation to previous results in small-scale repositories
- Limited performance metrics in benchmarking (could be included F1-score, confusion matrices, etc.)

**Requested Changes:**

A few proposed adjustments that could strengthen the work:
1. There are different level of information when describing the datasets; some of the dataset descriptions has proper information, but some does not. E.g, each dataset should include, at least, sampling frequency (or time steps) in proper units (not just the length of the sequences). These could be also added to Table 1. Furthermore, in image based datasets, size of the images and spatial resolution (in real units) should be given, and in the case of multi-sensor data, the name and type of sensors used.
2. There could be a common description of each six dataset categories (or using some other categorisation) that gives information about the temporal/time-series characteristics of the datasets, e.g., what are the typical challenges with these type of data from the model developer/practitioner perspective. Also, the input data types (e.g., counts, real values, pixel sets) could be discussed more carefully in each dataset category.
3. There could be a more comprehensive link to previous small-scale datasets in the benchmarking, considering analysis of bias-variance trade-off.
4. More versatile performance metrics in the benchmark could support the finding better, e.g., calculating F1-score (or precision and recall) and/or example confusion matrices in the case of unbalanced datasets.
5. It should be double-checked that randomly split datasets do not include very correlated examples in train and test sets (most of the cross-validation sets should be fine, though).

**Strengths And Weaknesses:**

Strengths
- Well-written and clearly structured manuscript
- There is a need for this kind of repository
- Proposal includes all the necessary information access to data and running the benchmarks

Weaknesses
- Few details missing from the dataset descriptions
- A bit limited experimental evidence comparing the proposed repository and previous ones
- Limited set of performance metrics utilised in the benchmark

---

### Review · Reviewer_bCAL · 2025-04-04

**Recommendation:** 3
**Confidence:** 2

**Summary Of Contributions:**

This paper presents MONSTER, a new dataset for studying time series classification. The dataset is presented as a complement to the existing UCR and UEA time series classification datasets. The main difference is that this dataset contains much larger time series, which can be more suitable to real world use cases. The authors conduct experiments on different time series algorithms on the new dataset.

**Strengths:**

See Strengths And Weaknesses

**Audience:**

Yes

**Claims And Evidence:**

Most of the claims are supported. One thing I am not really convinced is that the authors claim that the existing datasets favor those "low-variance" methods in section 2, however, it is not clear if the new dataset also has this problem or not. Also, It is interesting to know how different algorithms compare on both new and old datasets..

**Datasets And Benchmarks:**

The authors discussed how the datasets are collected and processed, the dataset is public in huggingface. However, there lacks a maintenance plan, although the authors mentioned that they aim to extend the datasets.

**Extended Submissions:**

N/A

**Limitations:**

See Strengths And Weaknesses

**Requested Changes:**

Weaknesses 1-3

**Strengths And Weaknesses:**

Strengths:
1. The curated dataset is a large dataset for time series classification, which aims to solve the problem that the existing repositories are a large set of small time series.
2. This paper provides meticulous empirical studies of existing time series methods on the new dataset.
3. The paper is well-written and easy to follow.

Weaknesses:
1. It would be nice to create a datasheet (https://www.microsoft.com/en-us/research/wp-content/uploads/2019/01/1803.09010.pdf) for this dataset, so that we have a better documentation for the dataset
2. The authors claim the limitation of the existing datasets and they favor "low variance methods", however, it is not clear how the new dataset solves this problem. It would be nice to include an experiment with low variance methods vs low bias models on old and new datasets. Also, it would be interesting to see if there is any algorithm that cannot be run on the new dataset.
3. It would be nice to include a hosting and maintenance plan

---

### Review · Reviewer_MP7L · 2025-04-06

**Recommendation:** 3
**Confidence:** 1

**Summary Of Contributions:**

The authors propose MONSTER, a collection of large-scale time series classification datasets to overcome the limitations of existing benchmarks, particularly: (1) small dataset size with a median of ~200. (2) failed to represent theoretical/practical challenges. (3) unrepresentative of the broader task of time series classifications, which implies the risk that SOTA models on existing benchmarks might fail to generalize to real-world time series classification problems. This might be particularly true when existing small datasets favor classifiers that minimize variance whereas larger datasets favor classifiers that reduce bias.

The MONSTER datasets overcome the above limitations by introducing 29 datasets with notably larger scales (from ~10k to ~59m time series) and various tasks (Audio, Satellite image time series, EEG, human activity recognition, counts, etc.). MONSTER datasets are evaluated with several deep learning models (FCN, TempCNN, etc.) and traditional models (Quant, Hydra, and ET) and show that 0-1 loss performance differs across dataset types and scales. The authors also discussed computation efficiency in section 4.4

**Strengths:**

As detailed in the Strengths And Weaknesses section.

**Audience:**

Yes

**Claims And Evidence:**

Yes.

**Datasets And Benchmarks:**

Yes.

**Extended Submissions:**

N/A

**Limitations:**

As detailed in the Strengths And Weaknesses section.

**Requested Changes:**

As detailed in the Weaknesses section.

**Strengths And Weaknesses:**

Strengths:
+ Identified and addressed an important limitation of existing time series benchmarks.
+ Propose large-scale MONSTER datasets covering a wide range of task categories.
+ Evaluate the MONSTER benchmark and discuss the preliminary results.


Weaknesses:

+ While computational efficiency is assessed, it’s somewhat less informative to me as the training time is evaluated on different GPUs with different compute throughputs. (i.e., A100 and A40). Ideally, it should evaluated on the same GPU spec for a fair comparison or at least report training FLOPs. Information on the software stack used for model training is also helpful.